# Depth Anything V2

**Lihe Yang[1]  Bingyi Kang[2†]  Zilong Huang[2]**
**Zhen Zhao  Xiaogang Xu  Jiashi Feng[2]  Hengshuang Zhao[1‡]**

[1]HKU    [2]TikTok
[†]project lead    [‡]corresponding author

https://depth-anything-v2.github.io

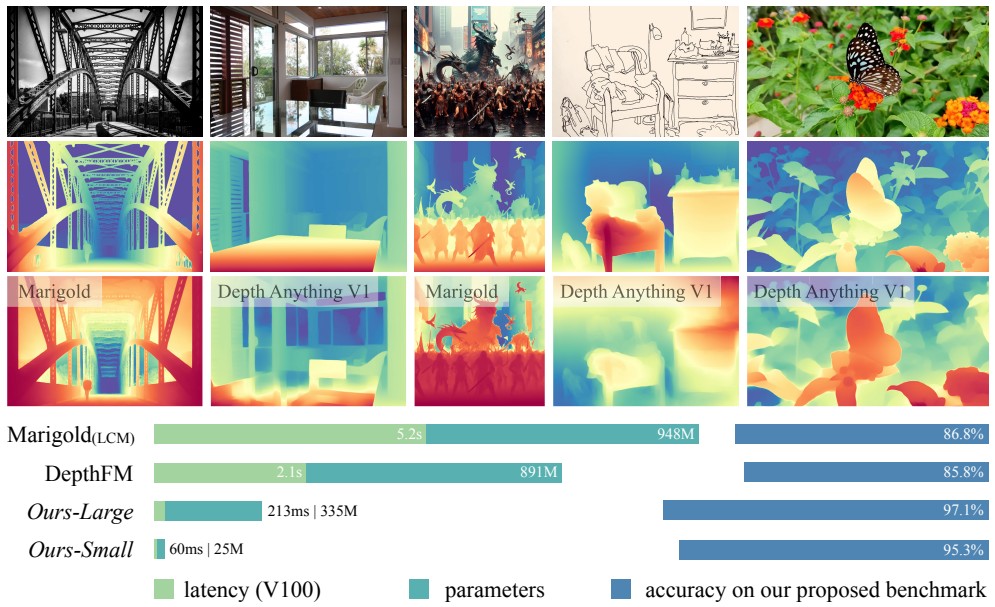

Figure 1: Depth Anything V2 significantly outperforms V1 [89] in robustness and fine-grained details. Compared with SD-based models [31, 25], it enjoys faster inference speed, fewer parameters, and higher depth accuracy.

## Abstract

This work presents *Depth Anything V2*. Without pursuing fancy techniques, we aim to reveal crucial findings to pave the way towards building a powerful monocular depth estimation model. Notably, compared with V1 [89], this version produces much finer and more robust depth predictions through three key practices: 1) replacing all labeled real images with synthetic images, 2) scaling up the capacity of our teacher model, and 3) teaching student models via the bridge of large-scale pseudo-labeled real images. Compared with the latest models [31] built on Stable Diffusion, our models are significantly more efficient (more than $10\times$ faster) and more accurate. We offer models of different scales (ranging from 25M to 1.3B params) to support extensive scenarios. Benefiting from their strong generalization capability, we fine-tune them with metric depth labels to obtain our metric depth models. In addition to our models, considering the limited diversity and frequent noise in current test sets, we construct a versatile evaluation benchmark with precise annotations and diverse scenes to facilitate future research.

Work done during an internship at TikTok.

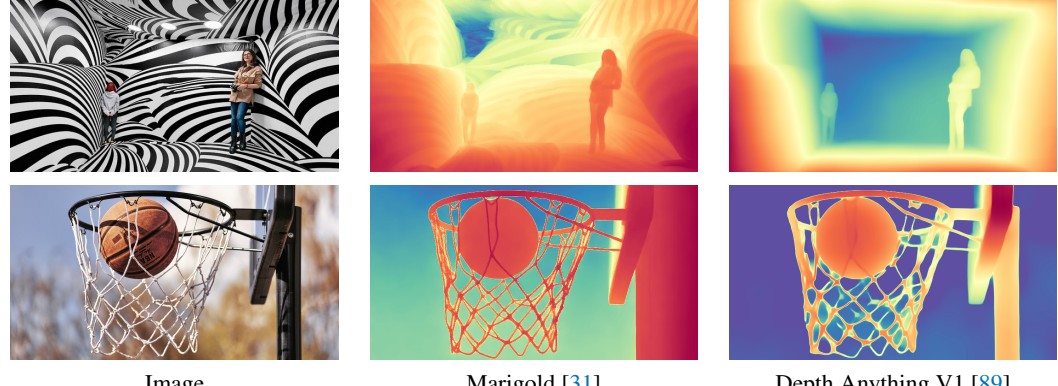

|          Image          |       Marigold [31]       |     Depth Anything V1 [89]     |

Figure 2: *Robustness* (1st row, the misleading room layout) of Depth Anything V1 and *Fine-grained detail* (2nd row, the thin basketball net) of Marigold.

| Preferable Properties | Fine Detail | Transparent Objects | Reflections | Complex Scenes | Efficiency | Transferability |
|---|---|---|---|---|---|---|
| Marigold [31] | ✓ | ✓ | ✓ | ✗ | ✗ | ✗ |
| Depth Anything V1 [89] | ✗ | ✗ | ✗ | ✓ | ✓ | ✓ |
| Depth Anything V2 (Ours) | ✓ | ✓ | ✓ | ✓ | ✓ | ✓ |

Table 1: Preferable properties of a powerful monocular depth estimation model.

# 1   Introduction

Monocular depth estimation (MDE) is gaining increasing attention, due to its fundamental role in widespread downstream tasks. Precise depth information is not only favorable in classical applications, such as 3D reconstruction [47, 32, 93], navigation [82], and autonomous driving [80], but is also preferable in modern scenarios, *e.g.*, AI-generated content, including images [101], videos [39], and 3D scenes [87, 64, 68]. Therefore, there have been numerous MDE models [56, 7, 6, 95, 26, 38, 31, 89, 88, 25, 20, 52, 28] emerging recently, which are all capable of addressing open-world images.

From the aspect of model architecture, these works can be divided into two groups. One group [7, 6, 89, 28] is based on discriminative models, *e.g.*, BEiT [4] and DINOv2 [50], while the other [31, 20, 25] is based on generative models, *e.g.*, Stable Diffusion (SD) [59]. In Figure 2, we compare two representative works from the two categories respectively: Depth Anything [89] as a discriminative model and Marigold [31] as a generative model. It can be easily observed that Marigold is superior in modeling the details, while Depth Anything produces more robust predictions for complex scenes. Moreover, as summarized in Table 1, Depth Anything is more efficient and lightweight than Marigold, with different scales to choose from. Meantime, however, Depth Anything is vulnerable to transparent objects and reflections, which are the strengths of Marigold.

In this work, taking all these factors into account, we aim to build a more capable foundation model for monocular depth estimation that can achieve all the strengths listed in Table 1:

- produce robust predictions for complex scenes, including but not limited to complex layouts, transparent objects (*e.g.*, glass), reflective surfaces (*e.g.*, mirrors, screens) [15], *etc.*
- contain fine details (comparable to the details of Marigold) in the predicted depth maps, including but not limited to thin objects (*e.g.*, chair legs) [42], small holes, *etc.*
- provide varied model scales and inference efficiency to support extensive applications [82].
- be generalizable enough to be transferred (*i.e.*, fine-tuned) to downstream tasks, *e.g.*, Depth Anything V1 serves as the pre-trained model for all the leading teams in the 3rd MDEC[1] [72].

Since the nature of MDE is a discriminative task, we start from Depth Anything V1 [89], aiming to maintain its strengths and rectify its weaknesses. Intriguingly, we will demonstrate that, to achieve

---

[1]https://jspenmar.github.io/MDEC

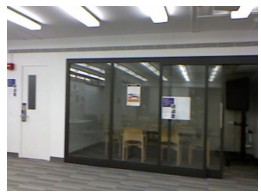 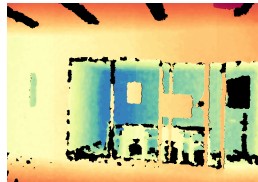 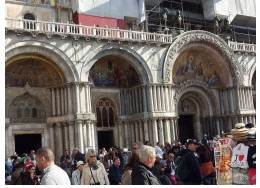 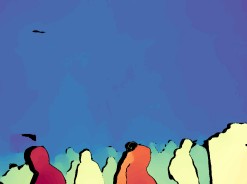

(a) Label noise in transparent object (depth sensor)  (b) Label noise in repetitive pattern (stereo matching)

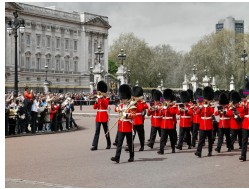 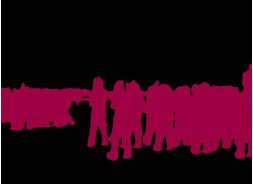 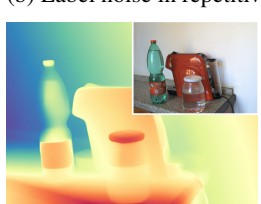 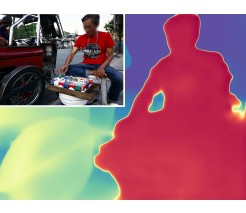

(c) Label noise in dynamic objects (SfM)  (d) Caused errors in model prediction

Figure 3: Various noise in "GT" depth labels (a: NYU-D [70], b: HRWSI [83], c: MegaDepth [37]) and prediction errors in correspondingly trained models (d). Black regions are ignored during training.

such a challenging goal, no fancy or sophisticated techniques need to be developed. The most critical part is still ***data***. It is indeed the same as the data-driven motivation of V1, which harnesses large-scale unlabeled data to speed up data scaling-up and increase the data coverage. In this work, we instead will first revisit its *labeled data* design, and then highlight the key role of unlabeled data.

We first present three key findings below. We will clarify them in detail in the following three sections.

**Q1 [Section 2]:** *Whether the coarse depth of MiDaS or Depth Anything come from the discriminative modeling itself? Is it a must to adopt the heavy diffusion-based modeling manner for fine details?*
**A1:** No, efficient discriminative models can also produce extremely fine details. The most critical modification is replacing all labeled real images with precise synthetic images.

**Q2 [Section 3]:** *Why do most prior works still stick to real images, if as A1 mentioned, synthetic images are already clearly superior to real images?*
**A2:** Synthetic images have their drawbacks, which are not trivial to address in previous paradigms.

**Q3 [Section 4]:** *How to avoid the drawbacks of synthetic images and also amplify its advantages?*
**A3:** Scale up the teacher model that is solely trained on synthetic images, and then teach (smaller) student models via the bridge of large-scale pseudo-labeled real images.

After the explorations, we successfully build a more capable MDE foundation model. However, we find current test sets [70] are too noisy to reflect the true strengths of MDE models. Thus, we further construct a versatile evaluation benchmark with precise annotations and diverse scenes (Section 6).

## 2 Revisiting the Labeled Data Design of Depth Anything V1

Building on the pioneering work of MiDaS [56, 7] in zero-shot MDE, recent studies tend to construct larger-scale training datasets in an effort to enhance estimation performance. Notably, Depth Anything V1 [89], Metric3D V1 [95] and V2 [28], as well as ZeroDepth [26], have amassed 1.5M, 8M, 16M, and 15M labeled images from various sources for training, respectively. However, few studies have critically examined this trend: *is such a huge amount of labeled images truly advantageous?*

Before answering it, let us first dig into the potentially overlooked drawbacks of ***real*** labeled images.

**Two disadvantages of real labeled data.** 1) *Label noise*, *i.e.*, *inaccurate labels* in depth maps. Stemming from the limitations inherent in various collection procedures, real labeled data inevitably contain inaccurate estimations. Such inaccuracies can arise from various factors, such as the inability of depth sensors to accurately capture the depth of transparent objects (Figure 3a), the vulnerability of stereo matching algorithms to textureless or repetitive patterns (Figure 3b), and the susceptible nature of SfM methods in handling dynamic objects or outliers (Figure 3c). 2) *Ignored details*. These real datasets often overlook certain details in their depth maps. As depicted in Figure 4a, the depth

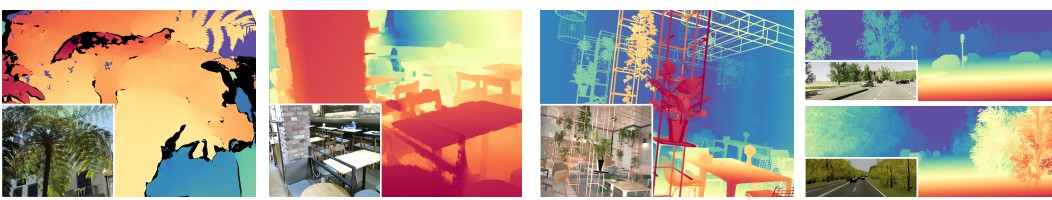

(a) Coarse depth of real data (HRWSI [83], DIML [14])  (b) Depth of synthetic data (Hypersim [58], vKITTI [9])

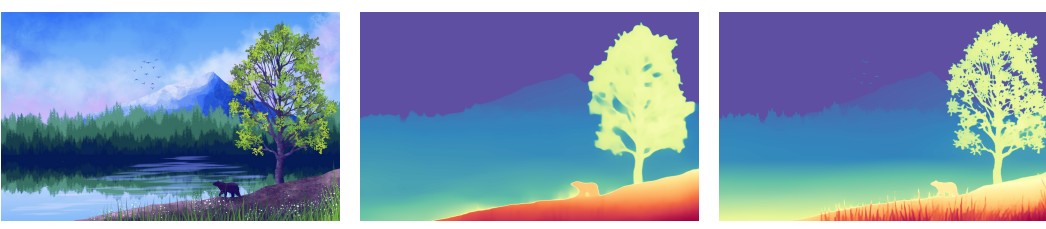

(c) Predictions of models trained on labeled real images (middle) and synthetic images (right)

Figure 4: Depth labels of real images (a) and synthetic images (b), and the corresponding model predictions (c). The labels of synthetic images are highly precise, and so are their trained models.

representation of the tree and chair is notably coarse. These datasets struggle to provide detailed supervision at object boundaries or within thin holes, resulting in over-smoothed depth predictions, as seen in the middle of Figure 4c. Hence, these noisy labels are so unreliable that the learned models make similar mistakes (Figure 3d). For example, MiDaS and Depth Anything V1 obtain poor scores of 25.9% and 53.5% respectively in the Transparent Surface Challenge [54] (more details in Table 12: our V2 achieves a competitive score of 83.6% in a zero-shot manner).

To overcome the above problems, we decide to change our training data and seek images with substantially better annotation. Inspired by several recent SD-based studies [31, 20, 25], that exclusively utilize synthetic images with complete depth information for training, we extensively check the label quality of synthetic images and note their potential to mitigate the drawbacks discussed above.

**Advantages of synthetic images.** Their depth labels are highly precise in two folds. 1) All fine details (*e.g.*, boundaries, thin holes, small objects, *etc.*) are correctly labeled. As demonstrated in Figure 4b, even all thin mesh structures and leaves are annotated with true depth. 2) We can obtain the actual depth of challenging transparent objects and reflective surfaces, *e.g.*, the vase on the table in Figure 4b. In a word, the depth of synthetic images is truly "GT". In the right side of Figure 4c, we show the fine-grained prediction of a MDE model trained on synthetic images. Moreover, we can quickly enlarge synthetic training images by collecting from graphics engines [58, 63, 53], which would not cause any privacy or ethical concerns, as compared to real images.

## 3 Challenges in Using Synthetic Data

If synthetic data are so advantageous, why are real data still dominating MDE? In this section, we identify **two limitations of synthetic images** that hinder them from being easily used in reality.

**Limitation 1.** There exists *distribution shift* between synthetic and real images. Although current graphics engines strive for photorealistic effects, their style and color distributions still evidently differ from real images. Synthetic images are too "clean" in color and "ordered" in layout, while real images contain more randomness. For instance, when comparing the images in Figure 4a and 4b, we can immediately distinguish the synthetic ones. Such distribution shift makes models struggle to transfer from synthetic to real images, even if the two data sources share similar layouts [57, 9].

**Limitation 2.** Synthetic images have *restricted scene coverage*. They are iteratively sampled from graphics engines with pre-defined fixed scene types, *e.g.*, "living room" and "street scene". Consequently, despite the astonishing precision of Hypersim [58] or Virtual KITTI [9] (Figure 4b), we cannot expect models trained on them to generalize well in real-world scenes like "crowded people". In contrast, some real datasets constructed from web stereo images (*e.g.*, HRWSI [83]) or monocular videos (*e.g.*, MegaDepth [37]), can cover extensive real-world scenes.

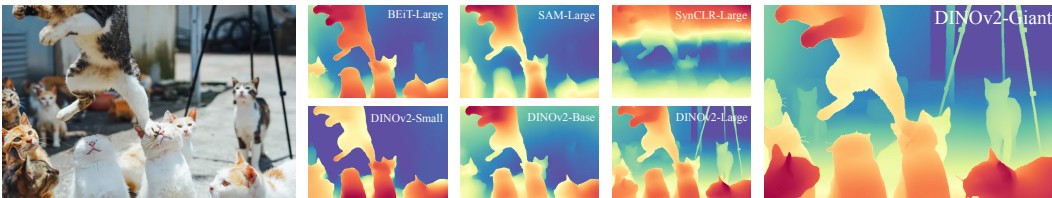

Figure 5: Qualitative comparison of different vision encoders on synthetic-to-real transfer. Only DINOv2-G produces a satisfying prediction. For quantitative comparisons, please refer to Section B.6.

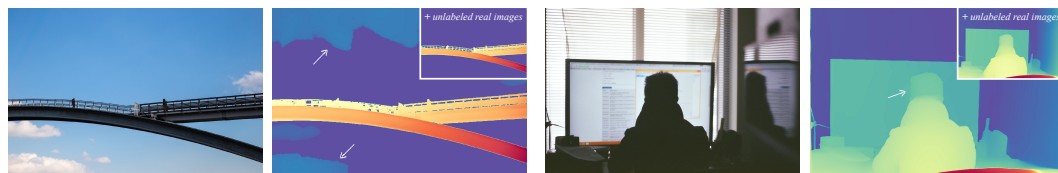

Figure 6: Failure cases of the most capable DINOv2-G model when purely trained on synthetic images. Left: the sky should be ultra far. Right: the depth of the head is not consistent with the body.

**Therefore, synthetic-to-real transfer is non-trivial in MDE.** To validate this claim, we conduct a pilot study to learning MDE models solely on synthetic images with four popular pre-trained encoders, including BEiT [4], SAM [33], SynCLR [75], and DINOv2 [50]. As illustrated in Figure 5, only DINOv2-G achieves satisfying results. All other model serials, as well as smaller DINOv2 models, suffer from severe generalization issues. This pilot study seems to give a straightforward solution to employing synthetic data in MDE, *i.e.*, building on the largest DINOv2 encoder, and relying on its inherent generalization ability. However, this naive solution faces two problems. First, DINOv2-G frequently encounters failure cases when the patterns of real test images are rarely presented in synthetic training images. In Figure 6, we can clearly observe incorrect depth predictions for the sky (cloud) and the human head. Such failures can be expected as our synthetic training sets do not include diverse sky patterns or humans. Moreover, most applications cannot accommodate the resource-intensive DINOv2-G model (1.3B) in terms of storage and inference efficiency. Actually, the smallest model in Depth Anything V1 is used most widely due to its real-time speed.

To alleviate the generalization issue, some works [7, 89, 28] use a combined training set of real and synthetic images. Unfortunately, as shown in Section B.9, the coarse depth map of real images is destructive to fine-grained prediction. Another potential solution is to collect more synthetic images, which is unsustainable as creating graphic engines mimicking every real-world scenario is intractable. Therefore, a reliable solution is demanding in building MDE models with synthetic data. In this paper, we will close this gap and present a roadmap that solves the preciseness and robustness dilemma *without any trade-offs*, and applicable to *any model scale*.

## 4 Key Role of Large-Scale Unlabeled Real Images

Our solution is straightforward: incorporating *unlabeled real* images. Our most capable MDE model, based on DINOv2-G, is initially trained purely on high-quality synthetic images. Then it assigns pseudo depth labels on unlabeled real images. Lastly, our new models are solely trained with large-scale and precisely pseudo-labeled images. Depth Anything V1 [89] has highlighted the importance of large-scale unlabeled real data. Here, in our special context of synthetic labeled images, we will demonstrate its *indispensable* role in more details from three perspectives.

**Bridge the domain gap.** As aforementioned, due to the distribution shift, directly transferring from synthetic training images to real test images is challenging. But if we can leverage extra real images as an intermediate learning target, the process will be more reliable. Intuitively, after explicitly training on pseudo-labeled real images, models can be more familiar with real-world data distribution. Compared with manually annotated images, our auto-generated pseudo labels are much more fine-grained and complete, as visualized in Figure 17.

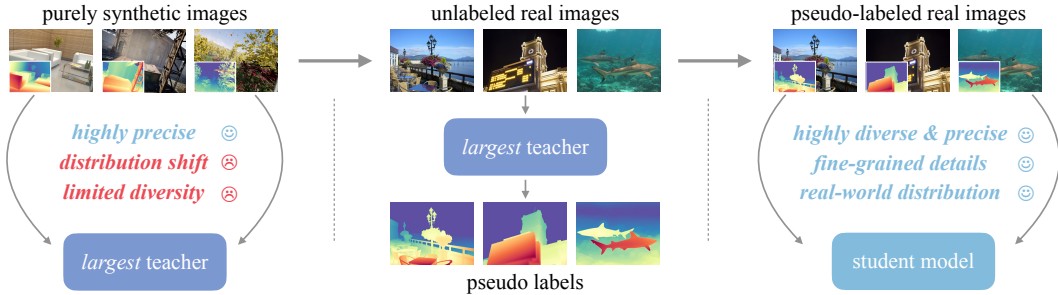

Figure 7: Depth Anything V2. We first train the most capable teacher on precise synthetic images. Then, to mitigate the distribution shift and limited diversity of synthetic data, we annotate unlabeled real images with the teacher. Finally, we train student models on high-quality pseudo-labeled images.

**Enhance the scene coverage.** The diversity of synthetic images is limited, without including enough real-world scenes. Nevertheless, we can easily cover numerous distinct scenes by incorporating large-scale unlabeled images from public datasets. Moreover, synthetic images are indeed very redundant due to being repetitively sampled from pre-defined videos. In comparison, unlabeled real images are clearly distinguished and very informative. By training on sufficient images and scenes, models not only demonstrate stronger zero-shot MDE capability (as shown in Figure 6 "+ *unlabeled real images*"), but they can also serve as better pre-trained sources for downstream related tasks [72].

**Transfer knowledge from the most capable model to smaller ones.** We have shown in Figure 5, that smaller models cannot directly benefit from synthetic-to-real transfer by themselves. However, armed with large-scale unlabeled real images, they can learn to mimic the high-quality predictions of the most capable model, similar to knowledge distillation [27]. But differently, our distillation is enforced at the label level via extra unlabeled real data, instead of at the feature or logit level with original labeled data. This practice is safer because there is evidence showing feature-level distillation is not always beneficial, especially when the teacher-student scale gap is huge [48]. Finally, as supported in Figure 16, unlabeled images boost the robustness of our smaller models tremendously.

## 5 Depth Anything V2

### 5.1 Overall Framework

According to all the above analysis, our final pipeline to train Depth Anything V2 is clear (Figure 7). It consists of three steps:

- train a reliable teacher model based on DINOv2-G *purely* on high-quality *synthetic* images.
- produce precise pseudo depth on large-scale unlabeled *real* images.
- train final student models on *pseudo-labeled real* images for robust generalization (we will show the synthetic images are not necessary in this step).

We will release four student models, based on DINOv2 small, base, large, and giant, respectively.

### 5.2 Details

As shown in Table 7, we use five precise synthetic datasets (595K images) and eight large-scale pseudo-labeled real datasets (62M images) for training. Same as V1 [89], for each pseudo-labeled sample, we ignore its top-$n$-largest-loss regions during training, where $n$ is set as 10%. We consider them as potentially noisy pseudo labels. Similarly, our models produce affine-invariant inverse depth[2]. We use two loss terms for optimization on labeled images: a scale- and shift-invariant loss $\mathcal{L}_{ssi}$ and a gradient matching loss $\mathcal{L}_{gm}$. These two objective functions are not new, as they are proposed by MiDaS [56]. But differently, we find $\mathcal{L}_{gm}$ is super beneficial to the depth sharpness when using synthetic images (Section B.7). On pseudo-labeled images, we follow V1 to add an additional feature alignment loss to preserve informative semantics from pre-trained DINOv2 encoders.

---

[2]To offer capable *metric depth* models, we further fine-tune our basic models with metric depth (Section 7.3).

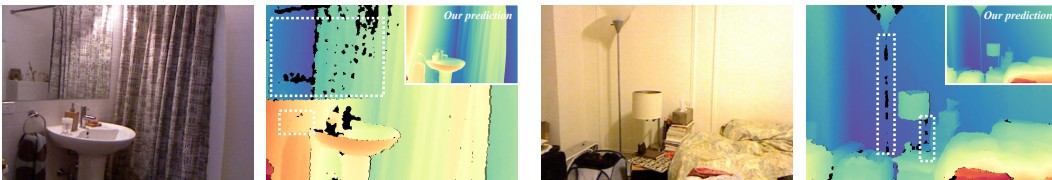

Figure 8: Visualization of widely adopted but indeed noisy test benchmark [70]. As highlighted, the depth of the mirror and thin structures are incorrect (black pixels are ignored). In comparison, our model predictions are accurate. The noise will cause better models instead achieve lower scores.

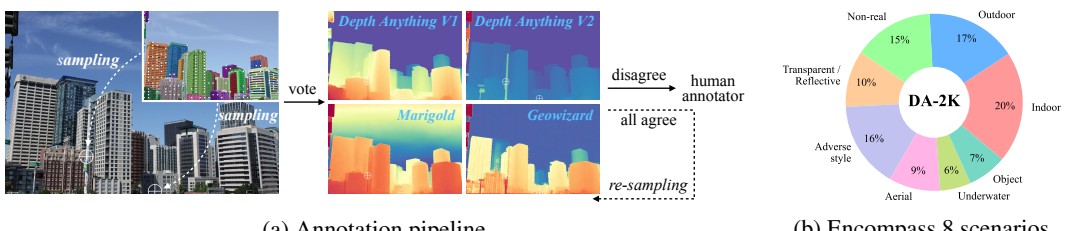

(a) Annotation pipeline      (b) Encompass 8 scenarios

Figure 9: Our proposed evaluation benchmark DA-2K. (a) The annotation pipeline for relative depth between two points. Points are sampled based on SAM [33] mask predictions. Disagreed pairs among four depth models will be popped out for annotators to label. (b) Detail of our scenario coverage.

## 6 A New Evaluation Benchmark: DA-2K

### 6.1 Limitations in Existing Benchmarks

In Section 2, we demonstrated that commonly used real training sets have noisy depth labels. Here, we further argue that widely adopted *test benchmarks* are also noisy. Figure 8 illustrates incorrect annotations for mirrors and thin structures on NYU-D [70] despite using specialized depth sensors. Such frequent label noise makes the reported metrics of powerful MDE models not reliable anymore.

Apart from label noise, another drawback of these benchmarks is *limited diversity*. Most of them were originally proposed for a single scene. For example, NYU-D [70] focuses on a few indoor rooms, while KITTI [24] only contains several street scenes. Performance on these benchmarks may not reflect real-world reliability. Ideally, we expect MDE models can handle any unseen scenes robustly.

The last problem in these existing benchmarks is *low resolution*. They mostly provide images with a resolution of around $500{\times}500$. But with modern cameras, we usually require precise depth estimation for higher-resolution images, *e.g.*, $1000{\times}2000$. It remains unclear whether the conclusions drawn from these low-resolution benchmarks can be safely transferred to high-resolution benchmarks.

### 6.2 DA-2K

Considering the above three limitations, we aim to construct a versatile evaluation benchmark for relative monocular depth estimation, that can 1) provide *precise* depth relationship, 2) cover *extensive* scenes, and 3) contain mostly *high-resolution* images for modern usage. Indeed, it is impractical for humans to annotate the depth of each pixel, especially for in-the-wild images. Thus, following DIW [11], we annotate *sparse* depth pairs for each image. Generally, given an image, we can select two pixels on it, and decide their relative depth between them (*i.e.*, which pixel is closer).

Concretely, we employ two distinct pipelines to select pixel pairs. In the first pipeline, as shown in Figure 9a, we use SAM [33] to automatically predict object masks. Instead of using the masks, we leverage key points (pixels) that prompt out them. We randomly sample two key pixels and query four expert models ([89, 31, 20] and ours) to vote on their relative depth. If there is disagreement, the pair will be sent to human annotators to decide the true relative depth. Due to potential ambiguity, annotators can skip any pair. However, there may be cases where all models incorrectly predict challenging pairs, and they are not flagged. To address this, we introduce a second pipeline, where we carefully analyze images and manually identify challenging pairs.

| Method | Encoder | KITTI [24] | | NYU-D [70] | | Sintel [8] | | ETH3D [62] | | DIODE [76] | |
|---|---|---|---|---|---|---|---|---|---|---|---|
| | | AbsRel | $\delta_1$ | AbsRel | $\delta_1$ | AbsRel | $\delta_1$ | AbsRel | $\delta_1$ | AbsRel | $\delta_1$ |
| MiDaS V3.1 [7] | ViT-L | 0.127 | 0.850 | 0.048 | 0.980 | 0.587 | 0.699 | 0.139 | 0.867 | 0.075 | 0.942 |
| Depth Anything V1 [89] | ViT-S | 0.080 | 0.936 | 0.053 | 0.972 | 0.464 | 0.739 | 0.127 | **0.885** | 0.076 | 0.939 |
| | ViT-B | 0.080 | 0.939 | 0.046 | 0.979 | **0.432** | 0.756 | **0.126** | 0.884 | 0.069 | 0.946 |
| | ViT-L | 0.076 | 0.947 | **0.043** | **0.981** | 0.458 | 0.760 | 0.127 | 0.882 | 0.066 | 0.952 |
| **Depth Anything V2** | ViT-S | 0.078 | 0.936 | 0.053 | 0.973 | 0.500 | 0.718 | 0.142 | 0.851 | 0.073 | 0.942 |
| | ViT-B | 0.078 | 0.939 | 0.049 | 0.976 | 0.495 | 0.734 | 0.137 | 0.858 | 0.068 | 0.950 |
| | ViT-L | **0.074** | 0.946 | 0.045 | 0.979 | 0.487 | 0.752 | 0.131 | 0.865 | 0.066 | 0.952 |
| | ViT-G | 0.075 | **0.948** | 0.044 | 0.979 | 0.506 | **0.772** | 0.132 | 0.862 | **0.065** | **0.954** |

Table 2: Zero-shot *relative* depth estimation. Better: AbsRel $\downarrow$, $\delta_1$ $\uparrow$. Solely from the metrics, Depth Anything V2 is better than MiDaS, but merely comparable with V1. But indeed, the focus and strengths of our V2 (*e.g.*, fine-grained details, robust to complex layouts, transparent objects, *etc.*) cannot be correctly reflected on these benchmarks. Similar results (*i.e.*, better model but worse score) are also observed in [7, 28].

| Method | Community Models | | | | Depth Anything V2 (Ours) | | | |
|---|---|---|---|---|---|---|---|---|
| | Marigold [31] | Geowizard [20] | DepthFM [25] | Depth Anything V1 [89] | ViT-S | ViT-B | ViT-L | ViT-G |
| Accuracy (%) | 86.8 | 88.1 | 85.8 | 88.5 | 95.3 | 97.0 | 97.1 | **97.4** |

Table 3: Performance on our proposed DA-2K evaluation benchmark, which encompasses eight representative scenarios. Even our most lightweight model is superior to all other community models.

To ensure preciseness, all annotations are triple-checked by the other two annotators. To ensure diversity, we first summarize eight important application scenarios of MDE (Figure 9b), and ask GPT-4 to produce diverse keywords related to each scenario. We then use these keywords to download corresponding images from Flickr. Finally, we annotate 1K images with 2K pixel pairs in total. Limited by space, please refer to Section C for details and comparisons with DIW [11].

**Position of DA-2K.** Despite the advantages, we *do not* expect DA-2K to *replace* current benchmarks. Accurate sparse depth is still far from the precise dense depth required for scene reconstruction. However, DA-2K can be considered a prerequisite for accurate dense depth. As such, we believe DA-2K can serve as *a valuable supplement* to existing benchmarks due to its extensive scene coverage and precision. It can also serve as a quick prior validation for users selecting community models for specific scenarios covered in DA-2K. Lastly, we believe it is also a potential testbed for the 3D awareness of future multimodal LLMs [41, 21, 3].

# 7 Experiment

## 7.1 Implementation details

Follow Depth Anything V1 [89], we use DPT [55] as our depth decoder, built on DINOv2 encoders. All images are trained at the resolution of 518×518 by resizing the shorter size to 518 followed by a random crop. When training the teacher model on synthetic images, we use a batch size of 64 for 160K iterations. In the third stage of training on pseudo-labeled real images, the model is trained with a batch size of 192 for 480K iterations. We use the Adam optimizer and set the learning rate of the encoder and the decoder as 5e-6 and 5e-5, respectively. In both training stages, we do not balance the training datasets, but simply concatenate them. The weight ratio of $\mathcal{L}_{ssi}$ and $\mathcal{L}_{gm}$ is set as 1:2.

## 7.2 Zero-Shot Relative Depth Estimation

**Performance on conventional benchmarks.** Since our model predicts affine-invariant *inverse* depth, for fairness, we compare with Depth Anything V1 [89] and MiDaS V3.1 [7] on five unseen test datasets. As shown in Table 2, our results are superior to MiDaS and comparable to V1 [89]. We are slightly inferior to V1 in *metrics* on two of the datasets. However, the plain metrics on these datasets are not the focus of this paper. This version aims to produce fine-grained predictions for thin structures and robust predictions for complex scenes, transparent objects, *etc.*. Improvement in these dimensions cannot be correctly reflected in current benchmarks.

**Performance on our proposed benchmark DA-2K.** As shown in Table 3, on our proposed benchmark with diverse scenes, even our smallest model is significantly better than other heavy SD-based

Table 4:

| Method | Higher is better ↑ | | | Lower is better ↓ | | | Method | Higher is better ↑ | | | Lower is better ↓ | | |
|---|---|---|---|---|---|---|---|---|---|---|---|---|---|
| | $\delta_1$ | $\delta_2$ | $\delta_3$ | AbsRel | RMSE | log10 | | $\delta_1$ | $\delta_2$ | $\delta_3$ | AbsRel | RMSE | RMSE log |
| AdaBins [5] | 0.903 | 0.984 | 0.997 | 0.103 | 0.364 | 0.044 | AdaBins [5] | 0.964 | 0.995 | 0.999 | 0.058 | 2.360 | 0.088 |
| DPT [55] | 0.904 | 0.988 | 0.998 | 0.110 | 0.357 | 0.045 | P3Depth [51] | 0.953 | 0.993 | 0.998 | 0.071 | 2.842 | 0.103 |
| P3Depth [51] | 0.898 | 0.981 | 0.996 | 0.104 | 0.356 | 0.043 | NeWCRFs [99] | 0.974 | 0.997 | 0.999 | 0.052 | 2.129 | 0.079 |
| SwinV2 [44] | 0.949 | 0.994 | 0.999 | 0.083 | 0.287 | 0.035 | SwinV2 [44] | 0.977 | 0.998 | 1.000 | 0.050 | 1.966 | 0.075 |
| AiT [49] | 0.954 | 0.994 | 0.999 | 0.076 | 0.275 | 0.033 | NDDepth [66] | 0.978 | 0.998 | 0.999 | 0.050 | 2.025 | 0.075 |
| VPD [102] | 0.964 | 0.995 | 0.999 | 0.069 | 0.254 | 0.030 | GEDepth [91] | 0.976 | 0.997 | 0.999 | 0.048 | 2.044 | 0.076 |
| IEBins [67] | 0.936 | 0.992 | 0.998 | 0.087 | 0.314 | 0.038 | IEBins [67] | 0.978 | 0.998 | 0.999 | 0.050 | 2.011 | 0.075 |
| ZoeDepth [6] | 0.951 | 0.994 | 0.999 | 0.077 | 0.282 | 0.033 | ZoeDepth [6] | 0.971 | 0.996 | 0.999 | 0.054 | 2.281 | 0.082 |
| Ours (ViT-S) | 0.961 | 0.996 | 0.999 | 0.073 | 0.261 | 0.032 | Ours (ViT-S) | 0.973 | 0.997 | 0.999 | 0.053 | 2.235 | 0.081 |
| Ours (ViT-B) | 0.977 | 0.997 | 1.000 | 0.063 | 0.228 | 0.027 | Ours (ViT-B) | 0.979 | 0.998 | 1.000 | 0.048 | 1.999 | 0.072 |
| Ours (ViT-L) | **0.984** | **0.998** | **1.000** | **0.056** | **0.206** | **0.024** | Ours (ViT-L) | **0.983** | **0.998** | **1.000** | **0.045** | **1.861** | **0.067** |

(a) NYU-D dataset      (b) KITTI dataset

Table 4: Fine-tuning our Depth Anything V2 pre-trained encoder to in-domain metric depth estimation, *i.e.*, training and test images share the same domain. All compared methods use the encoder size close to ViT-L.

| Encoder | $\mathcal{D}^l$ | $\mathcal{D}^u$ | KITTI [24] | | NYU-D [70] | | Sintel [8] | | ETH3D [62] | | DIODE [76] | | DA-2K |
|---|---|---|---|---|---|---|---|---|---|---|---|---|---|
| | | | AbsRel | $\delta_1$ | AbsRel | $\delta_1$ | AbsRel | $\delta_1$ | AbsRel | $\delta_1$ | AbsRel | $\delta_1$ | Acc (%) |
| ViT-S | ✓ | | 0.104 | 0.889 | 0.084 | 0.928 | 0.518 | 0.702 | 0.155 | 0.827 | 0.087 | 0.926 | 89.8 |
| | ✓ | ✓ | 0.085 | 0.928 | 0.054 | 0.971 | **0.491** | **0.723** | 0.143 | 0.849 | 0.074 | 0.941 | 94.1 |
| | | ✓ | **0.078** | **0.936** | **0.053** | **0.973** | 0.500 | 0.718 | **0.142** | **0.851** | **0.073** | **0.942** | **95.3** |
| ViT-B | ✓ | | 0.094 | 0.912 | 0.062 | 0.963 | 0.618 | 0.715 | 0.148 | 0.842 | 0.076 | 0.940 | 92.9 |
| | ✓ | ✓ | 0.080 | 0.938 | **0.049** | **0.976** | 0.515 | 0.732 | **0.137** | **0.859** | **0.068** | **0.950** | 96.7 |
| | | ✓ | **0.078** | **0.939** | **0.049** | **0.976** | **0.495** | **0.734** | **0.137** | 0.858 | **0.068** | **0.950** | **97.0** |
| ViT-L | ✓ | | 0.081 | 0.937 | 0.048 | 0.976 | 0.516 | 0.731 | 0.133 | 0.864 | 0.071 | 0.949 | 96.0 |
| | ✓ | ✓ | 0.075 | **0.947** | **0.045** | **0.979** | 0.542 | 0.741 | **0.130** | **0.866** | **0.066** | **0.953** | **97.3** |
| | | ✓ | **0.074** | 0.946 | **0.045** | **0.979** | **0.487** | **0.752** | 0.131 | 0.865 | **0.066** | 0.952 | 97.1 |
| Teacher model (ViT-G) | | | 0.075 | 0.947 | 0.044 | 0.979 | 0.530 | 0.767 | 0.131 | 0.865 | 0.066 | 0.954 | 97.4 |

Table 5: Importance of pseudo-labeled (unlabeled) real images ($\mathcal{D}^u$). $\mathcal{D}^l$: precisely labeled synthetic images.

models, *e.g.*, Marigold [31] and Geowizard [20]. Our most capable model achieves 10.6% higher accuracy than Margold in terms of relative depth discrimination. Please refer to Table 14 for the comprehensive per-scenario performance of our models.

## 7.3 Fine-tuned to Metric Depth Estimation

To validate the generalization ability of our model, we transfer its encoder to the downstream metric depth estimation task. First, same as V1 [89], we follow the ZoeDepth [6] pipeline, but replace its MiDaS [7] encoder with our pre-trained encoder. As shown in Table 4, we achieve significant improvements over previous methods on both NYU-D and KITTI datasets. Notably, even our most lightweight model which is based on ViT-S, is superior to other models built on ViT-L [6].

Although the reported metrics look impressive, models trained on NYUv2 or KITTI fail to produce fine-grained depth prediction and are not robust to transparent objects, due to the inherent noise in training sets. Therefore, to satisfy real-world applications such as multi-view synthesis, we fine-tune our powerful encoder on Hypersim [58] and Virtual KITTI [9] synthetic datasets, for indoor and outdoor metric depth estimation, respectively. We will release these two metric depth models. Please refer to Figure 15 for qualitative comparisons with the previous ZoeDepth method.

## 7.4 Ablation Study

Limited by space, we defer most of our ablations to the appendix except for two on pseudo labels.

**Importance of large-scale pseudo-labeled real images.** As shown in Table 5, compared with solely trained on synthetic images, our models are greatly enhanced by incorporating pseudo-labeled real images. Different from Depth Anything V1 [89], we further attempt to remove the synthetic images during training student models. We find this can even lead to slightly better results for smaller models (*e.g.*, ViT-S and ViT-B). So we finally choose to train student models purely on pseudo-labeled images. This observation is indeed similar to SAM [33] that only releases its pseudo-labeled masks.

| Label Source | KITTI [24] | | NYU-D [70] | | Sintel [8] | | ETH3D [62] | | DIODE [76] | | DA-2K |
|---|---|---|---|---|---|---|---|---|---|---|---|
| | AbsRel | $\delta_1$ | AbsRel | $\delta_1$ | AbsRel | $\delta_1$ | AbsRel | $\delta_1$ | AbsRel | $\delta_1$ | Acc (%) |
| Manual Label | 0.122 | 0.882 | 0.074 | 0.952 | 0.581 | 0.693 | 0.159 | 0.832 | 0.126 | 0.890 | 80.2 |
| Pseudo Label | **0.099** | **0.901** | **0.062** | **0.963** | **0.514** | **0.701** | **0.147** | **0.843** | **0.084** | **0.929** | **89.7** |

Table 6: Comparison between originally manual label and our produced pseudo label on the DIML dataset [14]. Our produced pseudo labels are of much higher quality than the manual labels provided by DIML.

**Pseudo label *vs*. manual label on real labeled images.** We have demonstrated before in Figure 4a that existing labeled real datasets are very noisy. Here we conduct a quantitative comparison. We use real images from the DIML [14] dataset, and compare the transferring performance under its original manual label and our produced pseudo label respectively. We can observe in Table 6 that the model trained with pseudo labels is significantly better than the manual-label counterpart. The huge gap indicates the high quality of our pseudo labels and the rich noise in current labeled real datasets.

## 8 Related Work

**Monocular depth estimation.** Early works [18, 19, 5] focus on the in-domain metric depth estimation, where training and test images must share the same domain [70, 24]. Due to their restricted application scenarios, recently there has been increasing attention on zero-shot relative monocular depth estimation. Among them, some works address this task through better modeling manners, *e.g.*, using Stable Diffusion [59] as a depth denoiser [31, 25, 20]. Other works [94, 96, 89] focus on the data-driven perspective. For example, MiDaS [56, 55, 7] and Metric3D [95] collect 2M and 8M labeled images respectively. Aware of the difficulty of scaling up labeled images, Depth Anything V1 [89] leverages 62M unlabeled images to enhance the model's robustness. In this work, differently, we point out multiple limitations in widely used labeled real images. We thus especially highlight the necessity of resorting to synthetic images to ensure depth preciseness. Meantime, to tackle the generalization issue caused by synthetic images, we adopt both data-driven (large-scale pseudo-labeled real images) and model-driven (scaling up the teacher model) strategies.

**Learning from unlabeled real images.** How to learn informative representations from unlabeled images is widely studied in the field of semi-supervised learning [36, 86, 71, 90]. However, they focus on academic benchmarks [34] which only allow usage of small-scale labeled and unlabeled images. In comparison, we study a real-world application scenario, *i.e.*, how to further boost the baseline of 0.6M labeled images with 62M unlabeled images. Moreover, distinguished from Depth Anything V1 [89], we exhibit the indispensable role of unlabeled real images especially when we replace all labeled real images with synthetic images [22, 23, 61]. We demonstrate "precise synthetic data + pseudo-labeled real data" is a more promising roadmap than labeled real data.

**Knowledge distillation.** We distill transferable knowledge from our most capable teacher model to smaller models. This is similar to the core spirit of knowledge distillation (KD) [27]. But we are also fundamentally different in that we perform distillation at the *prediction level* through extra *unlabeled* real images, while KD [2, 73, 100] typically studies better distillation strategies at the *feature or logit* level through *labeled* images. We aim to reveal the importance of large-scale unlabeled data and larger teacher model, rather than delicate loss designs [43, 69] or distillation pipelines [10]. Moreover, it is indeed non-trivial and risky to directly distill feature representations between two models with a tremendous scale gap [48]. In comparison, our pseudo-label distillation is easier and safer.

## 9 Conclusion

In this work, we present *Depth Anything V2*, a more powerful foundation model for monocular depth estimation. It is capable of 1) providing robust and fine-grained depth prediction, 2) supporting extensive applications with varied model sizes (from 25M to 1.3B parameters), and 3) being easily fine-tuned to downstream tasks as a promising model initialization. We reveal crucial findings to pave the way towards building a strong MDE model. Furthermore, realizing the poor diversity and rich noise in existing test sets, we construct a versatile evaluation benchmark DA-2K, covering diverse high-resolution images with precise and challenging sparse depth labels.

**Acknowledgment.** This work is supported by the National Natural Science Foundation of China (No.62201484), HKU Startup Fund, and HKU Seed Fund for Basic Research.

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

# Appendix

For a thorough understanding and visualization of our Depth Anything V2, we compile a comprehensive appendix. The following table of contents will direct you to specific sections of interest.

## Contents

## A  Sources of Training Data

As listed in Table 7, we replace all labeled real datasets in Depth Anything V1 [89] with five synthetic datasets for label preciseness. Then, to mitigate the issues of distribution shift and limited diversity caused by synthetic images, we further leverage eight large-scale public datasets, comprising 62M real images with great diversity. We only use their raw images, and assign depth to them with our most capable teacher model. Student models are trained purely on these pseudo-labeled real images.

## B  Experiments

### B.1  Fine-tuned to semantic segmentation

Similar to the practice in metric MDE, we further fine-tune our pre-trained encoder to downstream semantic segmentation task to especially examine its semantic awareness. As demonstrated in Table 8,

| Dataset | Indoor | Outdoor | # Images |
|---|---|---|---|
| Precise *Synthetic* Images (595K) | | | |
| BlendedMVS [92] | ✓ | ✓ | 115K |
| Hypersim [58] | ✓ | | 60K |
| IRS [77] | ✓ | | 103K |
| TartanAir [79] | ✓ | ✓ | 306K |
| VKITTI 2 [9] | | ✓ | 20K |
| Pseudo-labeled *Real* Images (62M) | | | |
| BDD100K [97] | | ✓ | 8.2M |
| Google Landmarks [81] | | ✓ | 4.1M |
| ImageNet-21K [60] | ✓ | ✓ | 13.1M |
| LSUN [98] | ✓ | | 9.8M |
| Objects365 [65] | ✓ | ✓ | 1.7M |
| Open Images V7 [35] | ✓ | ✓ | 7.8M |
| Places365 [103] | ✓ | ✓ | 6.5M |
| SA-1B [33] | ✓ | ✓ | 11.1M |

Table 7: Our training data sources.

| Method | Encoder | mIoU |
|---|---|---|
| DDP [30] | Swin-S [45] | 82.4 |
| Depth Anything V2 | Small | **82.9** |
| DDP [30] | Swin-B [45] | 82.5 |
| Depth Anything V2 | Base | **83.9** |
| Segmenter [74] | ViT-L [17] | 82.2 |
| SegFormer [85] | MiT-B5 [85] | 82.4 |
| Mask2Former [13] | Swin-L [45] | 83.3 |
| OneFormer [29] | Swin-L [45] | 83.0 |
| OneFormer [29] | ConvNeXt-XL [46] | 83.6 |
| DDP [30] | ConvNeXt-L [46] | 83.2 |
| Depth Anything V2 | Large | **85.6** |

(a) Cityscapes dataset

| Method | Encoder | mIoU |
|---|---|---|
| UperNet [84] | InternImage-S [78] | 50.1 |
| Depth Anything V2 | Small | **53.9** |
| UperNet [84] | InternImage-B [78] | 50.8 |
| Depth Anything V2 | Base | **57.1** |
| UperNet [84] | InternImage-XL [78] | 55.0 |
| UperNet [84] | BEiT-L [4] | 56.3 |
| Mask2Former [13] | Swin-L [45] | 56.4 |
| ViT-Adapter [12] | BEiT-L [4] | 58.3 |
| OneFormer [29] | Swin-L [45] | 57.4 |
| OneFormer [29] | ConNeXt-XL [46] | 57.4 |
| Depth Anything V2 | Large | **58.6** |

(b) ADE20K dataset

Table 8: Transferring our Depth Anything V2 encoders to semantic segmentation. We adopt Mask2Former as our segmentation model. We achieve the results *without* Mapillary [1] or COCO [40] pre-training.

our models of various scales consistently achieve the best performance, outperforming other methods remarkably. These promising results indicate the potential of our model to serve as the initialization for diverse downstream semantic-related tasks.

## B.2 Transferring performance of each *labeled* dataset

We totally use five synthetic datasets to train our teacher model for pseudo labeling. Here we examine their individual effect on the model generalization capability. As demonstrated in Table 9, among them, the two purely indoor datasets Hypersim [58] and IRS [77] surprisingly fuel the most generalization ability. Although VKITTI 2 [9] has poor metric results, we find it is highly beneficial to the prediction sharpness, due to the large number of fine-grained structures (*e.g.*, leaves) in its training samples. Moreover, BlendedMVS [92] is critical to the capability of dealing with the bird's-eye view. Overall, each dataset has its own good properties to benefit the combined performance.

| Labeled Dataset | KITTI [24] | | NYU-D [70] | | Sintel [8] | | ETH3D [62] | | DIODE [76] | |
|---|---|---|---|---|---|---|---|---|---|---|
| | AbsRel | $\delta_1$ | AbsRel | $\delta_1$ | AbsRel | $\delta_1$ | AbsRel | $\delta_1$ | AbsRel | $\delta_1$ |
| BlendedMVS [92] | 0.088 | 0.919 | 0.069 | 0.957 | 0.538 | 0.661 | 0.150 | 0.839 | 0.095 | 0.915 |
| Hypersim [58] | 0.086 | 0.928 | 0.054 | 0.972 | 0.550 | 0.711 | **0.123** | **0.884** | 0.088 | 0.937 |
| IRS [77] | 0.100 | 0.900 | 0.055 | 0.973 | **0.435** | **0.738** | 0.149 | 0.831 | 0.084 | 0.931 |
| TartanAir [79] | 0.094 | 0.913 | 0.063 | 0.963 | 0.618 | 0.710 | 0.159 | 0.820 | 0.088 | 0.929 |
| VKITTI 2 [9] | 0.102 | 0.896 | 0.127 | 0.842 | 0.887 | 0.663 | 0.215 | 0.714 | 0.134 | 0.867 |
| All labeled data | **0.081** | **0.937** | **0.048** | **0.976** | 0.516 | 0.731 | 0.133 | 0.864 | **0.071** | **0.949** |

Table 9: Transferring performance of each *labeled* dataset with ViT-L. **Best results**, second best results.

## B.3 Transferring performance of each *unlabeled* dataset

We further analyze the benefit of each unlabeled source in Table 10. Accordingly, we present three observations. 1) Except the Sintel [8] synthetic game test set, unlabeled real images benefit all test sets tremendously. 2) When unlabeled images and test images share the same domain, the test results are improved most, *e.g.*, LSUN (indoor) improves the $\delta_1$ metric on NYU-D (indoor) from 0.928 $\rightarrow$ 0.970. 3) Even if unlabeled images and test images belong to contradictory domains, unlabeled images are still beneficial, *e.g.*, LSUN improves the $\delta_1$ on KITTI (street scene) from 0.889 $\rightarrow$ 0.913.

| Dataset | KITTI [24] | | NYU-D [70] | | Sintel [8] | | ETH3D [62] | | DIODE [76] | |
|---|---|---|---|---|---|---|---|---|---|---|
| | AbsRel | $\delta_1$ | AbsRel | $\delta_1$ | AbsRel | $\delta_1$ | AbsRel | $\delta_1$ | AbsRel | $\delta_1$ |
| Labeled datasets | 0.104 | 0.889 | 0.084 | 0.928 | 0.518 | 0.702 | 0.155 | 0.827 | 0.087 | 0.926 |
| + BDD100K | 0.091 | 0.916 | 0.071 | 0.951 | 0.600 | 0.708 | 0.153 | 0.834 | 0.087 | 0.927 |
| + Google Landmarks | 0.091 | 0.918 | 0.063 | 0.963 | 0.566 | 0.704 | 0.145 | 0.844 | 0.078 | 0.938 |
| + ImageNet-21K | 0.089 | 0.923 | 0.060 | 0.965 | 0.579 | 0.703 | 0.148 | 0.840 | 0.083 | 0.932 |
| + LSUN | 0.093 | 0.913 | 0.055 | 0.970 | 0.529 | 0.707 | 0.148 | 0.839 | 0.084 | 0.931 |
| + Objects365 | 0.089 | 0.920 | 0.058 | 0.967 | 0.551 | 0.701 | 0.145 | 0.846 | 0.080 | 0.937 |
| + Open Images V7 | 0.089 | 0.921 | 0.060 | 0.965 | 0.606 | 0.712 | 0.144 | 0.847 | 0.080 | 0.937 |
| + Places365 | 0.090 | 0.919 | 0.059 | 0.967 | 0.539 | 0.705 | 0.150 | 0.839 | 0.080 | 0.937 |
| + SA-1B | 0.092 | 0.915 | 0.067 | 0.956 | 0.652 | 0.708 | **0.142** | **0.850** | 0.080 | 0.935 |
| + All unlabeled data | **0.085** | **0.928** | **0.054** | **0.971** | **0.491** | **0.723** | 0.143 | 0.849 | **0.074** | **0.941** |

Table 10: Transferring performance by incorporating each *unlabeled* dataset with ViT-S. **Best**, second best.

## B.4 Are such large-scale unlabeled images really necessary?

We have proved that our used 62M unlabeled images are critical to model performance. However, we question that, is such a huge scale really necessary? What if we only use part of unlabeled sets and iterate the model for more epochs on it? To validate this, we solely use the SA-1B [33] dataset as our unlabeled source and train a model on it for the same iterations we use for 62M unlabeled images. As shown in Table 11, data diversity (*i.e.*, more datasets) is still highly important, which cannot be bridged by simply iterating a single dataset for more cycles. So we believe our large-scale unlabeled real images are necessary to ensure open-world generalization.

| Unlabeled Sets | # Images | Iterations | KITTI [24] | | NYU-D [70] | | Sintel [8] | | ETH3D [62] | | DIODE [76] | |
|---|---|---|---|---|---|---|---|---|---|---|---|---|
| | | | AbsRel | $\delta_1$ | AbsRel | $\delta_1$ | AbsRel | $\delta_1$ | AbsRel | $\delta_1$ | AbsRel | $\delta_1$ |
| SA-1B [33] | 11M | 480K | 0.090 | 0.915 | 0.073 | 0.948 | 0.588 | 0.707 | **0.141** | **0.852** | 0.073 | 0.942 |
| All eight sets | 62M | | **0.085** | **0.928** | **0.054** | **0.971** | **0.491** | **0.723** | 0.143 | 0.849 | 0.074 | 0.941 |

Table 11: Training the model solely on SA-1B for the same iterations as all sets (thus more cycles) with ViT-S.

## B.5 Performance on transparent or reflective surfaces

As aforementioned, one advantage of synthetic samples is the precise depth of the challenging transparent and reflective surfaces, which is important in navigation applications [82]. To validate the performance of our V2 in this specific domain, we compare different model predictions in the latest NTIRE 2024 Transparent Surface Challenge[3] [54]. Validation results are summarized in Table 12. Our V2 model achieves a remarkable boost over MiDaS [56] and Depth Anything V1 [89] in a zero-shot manner. Further, by simply fine-tuning our model with the challenge training data, we can nearly achieve the first-place score (0.912 *vs*. 0.917). Compared with the DINOv2 [50] encoder, our pre-trained model acts as a much stronger initialization (0.758 *vs*. 0.912).

| Method | Zero-shot (no fine-tuning) | | | Simple fine-tuning | | First place |
|---|---|---|---|---|---|---|
| | MiDaS V3.1 [7] | Depth Anything V1 [89] | V2 (Ours) | DINOv2 [50] | Depth Anything V2 (Ours) | |
| $\delta_1$ ($\uparrow$) | 0.259 | 0.535 | 0.836 | 0.758 | **0.912** | **0.917** |

Table 12: Results under different models and strategies in the NTIRE 2024 Transparent Surface Challenge [54].

---

[3] https://cvlab-unibo.github.io/booster-web/ntire24.html

## B.6    Comparison among various pre-trained encoders

We compare several currently most powerful pre-trained encoders in our MDE task, including BEiT [4], SAM [33], SynCLR [75], DINOv2 [50], and DINOv2 with registers [16]. As shown in Table 13, at the ViT-large scale, we find DINOv2 serial [50, 16] is remarkably superior to all other encoders. The success of DINOv2 further reflects the promising future of the data-driven roadmap, since it carefully collects 142M pre-training data without designing fancy algorithms or architectures.

When scaling up the ViT-large encoder to ViT-giant, we surprisingly observe DINOv2-G Reg [16] is much inferior to the non-register initial version [50]. This is the same as the findings in Probe3D [3]. Thus, we choose to build our teacher and student models on the original DINOv2 encoders.

| Encoder | KITTI [24] | | NYU-D [70] | | Sintel [8] | | ETH3D [62] | | DIODE [76] | |
|---|---|---|---|---|---|---|---|---|---|---|
| | AbsRel | $\delta_1$ | AbsRel | $\delta_1$ | AbsRel | $\delta_1$ | AbsRel | $\delta_1$ | AbsRel | $\delta_1$ |
| BEiT-L [4] | 0.149 | 0.814 | 0.068 | 0.950 | 0.777 | 0.627 | 0.145 | 0.846 | 0.103 | 0.912 |
| SAM-L [33] | 0.104 | 0.893 | 0.186 | 0.745 | 0.703 | 0.688 | 0.143 | 0.849 | 0.108 | 0.907 |
| SynCLR-L [75] | 0.278 | 0.650 | 0.344 | 0.469 | 1.608 | 0.493 | 0.301 | 0.638 | 0.262 | 0.712 |
| DINOv2-L [50] | 0.081 | 0.937 | **0.048** | **0.976** | **0.516** | 0.731 | **0.133** | **0.864** | 0.071 | 0.949 |
| DINOv2-L Reg [16] | **0.078** | **0.942** | 0.049 | 0.975 | 0.522 | **0.734** | 0.138 | 0.856 | **0.068** | **0.952** |
| DINOv2-G [50] | **0.075** | **0.947** | **0.044** | **0.979** | **0.530** | **0.767** | **0.131** | **0.865** | **0.066** | **0.954** |
| DINOv2-G Reg [16] | 0.084 | 0.926 | 0.061 | 0.964 | 0.753 | 0.729 | 0.141 | 0.852 | 0.086 | 0.931 |

Table 13: Comparison among various pre-trained encoders when purely trained on synthetic images.

## B.7    Benefit of gradient matching loss to fine-grained predictions

MiDaS [56] proposes a gradient matching loss $\mathcal{L}_{gm}$ to enhance the depth sharpness. Unfortunately, we find this loss term fails to bring evident improvement when the model is trained on labeled real datasets. We speculate that, the *sparse* and *coarse* groundtruth label in real datasets cannot provide fine-grained supervision, even with this explicit regularization. To check this, we further apply and ablate this loss term on synthetic training datasets, whose labels are complete and highly precise. We gradually increase the loss weight of $\mathcal{L}_{gm}$ and observe the corresponding depth sharpness. As shown in Figure 10, when the weight is increased from the default 0.5 to 4.0, the sharpness is steadily improved. We finally set the weight as 2.0 to trade off between the metric results and sharpness.

| Image | Loss weight 0.5 | Loss weight 2.0 | Loss weight 4.0 |
|---|---|---|---|

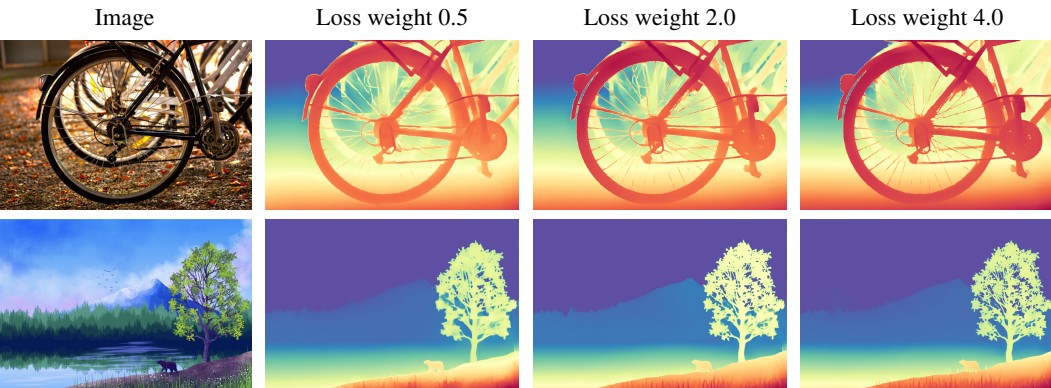

Figure 10: Effect of the gradient matching loss $\mathcal{L}_{gm}$ in terms of fine-grained details.

## B.8    Test-time resolution scaling up

By default, we test images at the same resolution as that used in training, *i.e.*, resizing the shorter size to 518 with the aspect ratio kept. This is a common practice to achieve the optimal performance. However, we surprisingly find that our model has the property of "test-time resolution scaling up". It means we can almost freely increase the image resolution at test time to produce more fine-grained depth maps. As shown in Figure 11, when gradually adjusting the resolution by $2\times$ and $4\times$ of the base resolution (518), the depth sharpness is also gradually improved.

| Image | 1x resolution | 2x resolution | 4x resolution |
|---|---|---|---|

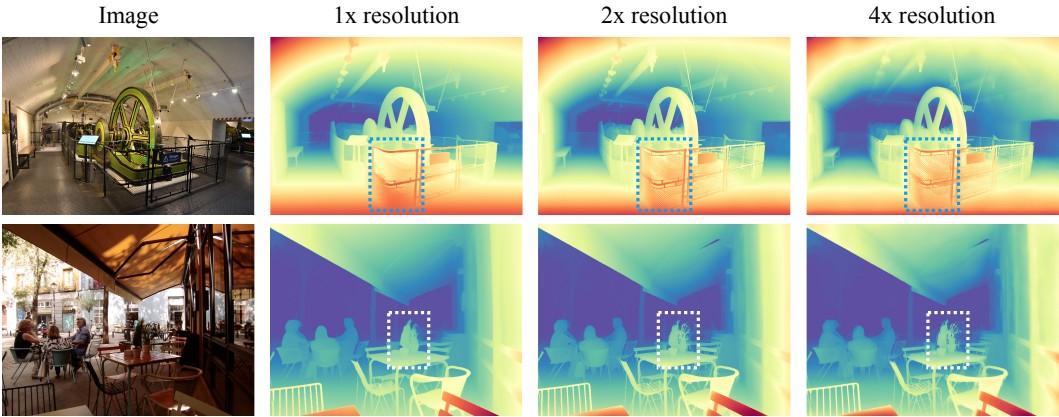

Figure 11: Test-time resolution scaling up can further improve the prediction sharpness.

## B.9 Harm of real labeled images to fine-grained predictions

According to the ablation study in Depth Anything V1 [89], HRWSI [83] is the best-performed real training dataset. We attempt to add it to our synthetic training sets. However, as shown in Figure 12, we find although it only accounts for 5% of the total training images, its coarse depth labels have a huge negative impact on the original fine-grained predictions. So we choose to use purely synthetic images to train our largest teacher model to ensure the supervision preciseness.

| Image | Purely synthetic images | Synthetic images + HRWSI |
|---|---|---|

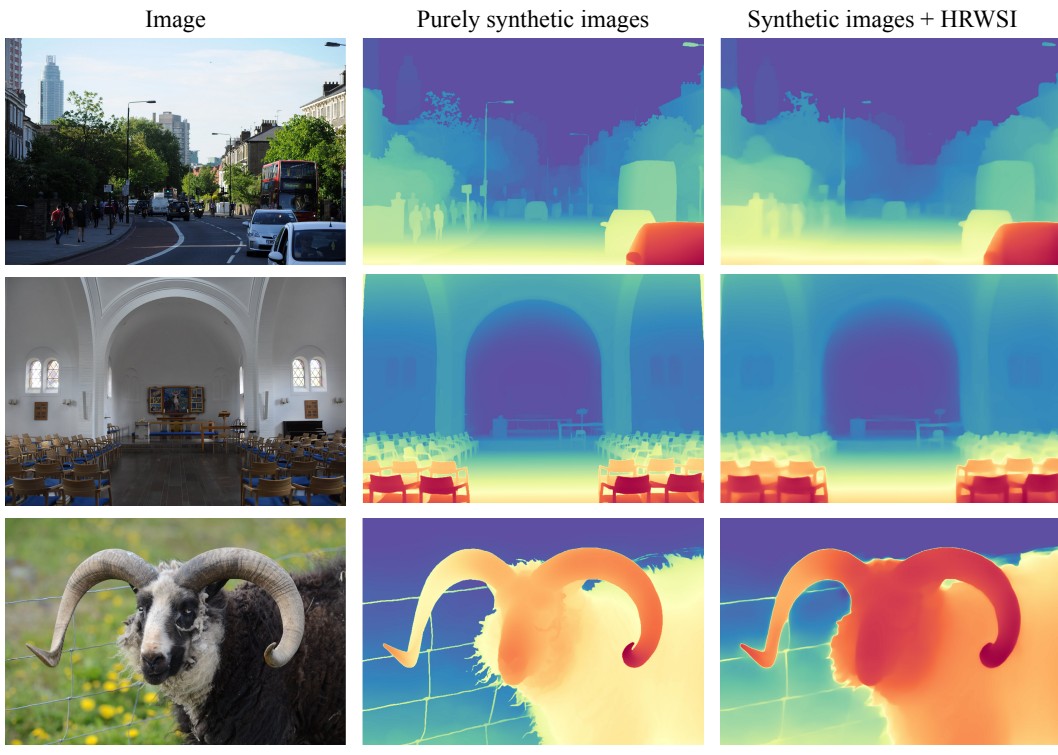

Figure 12: Adding real training dataset, *e.g.*, HRWSI, to synthetic training datasets, will ruin the original fine-grained depth predictions.

## B.10 Qualitative comparison between Depth Anything V1 and V2

Please refer to Figure 13. Our Depth Anything V2 produces much more fine-grained depth predictions than V1 [89]. Ours are also highly robust to transparent objects.

### B.11   Qualitative comparison between Marigold and Depth Anything V2

Please refer to Figure 14. Our Depth Anything V2 is significantly more robust than Marigold [31].

### B.12   Qualitative comparison between our metric depth models and ZoeDepth

We fine-tune our finally released metric depth models purely on synthetic datasets, such as Hypersim [58] and Virtual KITTI [9]. In Figure 15, we compare our metric depth predictions with ZoeDepth, which is trained on real datasets like NYUv2 [70].

### B.13   Qualitative comparison between w/ and w/o pseudo-labeled real images

Please refer to Figure 16. As shown, purely trained on precise synthetic images, the DINOv2-small-based model suffers severe generalization problem. However, when trained on the high-quality and diverse pseudo-labeled real images, even the small model (25M parameters) exhibits powerful generalization capability to complex scenes.

### B.14   Qualitative results of produced pseudo labels

Please refer to Figure 17. Our teacher produces highly precise pseudo labels on diverse real images.

### B.15   Qualitative results on test benchmarks

Please refer to Figure 18. Our model is consistently better than V1 [89] on standard benchmarks.

## C   DA-2K Evaluation Benchmark

### C.1   Per-scenario accuracy

We report the per-scenario accuracy on our DA-2K evaluation benchmark. By comparing the results of training on labeled synthetic images ($\mathcal{D}^l$) and pseudo-labeled real images ($\mathcal{D}^u$), we can clearly see the value of large-scale unlabeled data and also the preciseness of our pseudo labels.

| Encoder | $\mathcal{D}^l$ | $\mathcal{L}^u$ | Indoor | Outdoor | Non-real | Transparent | Adverse style | Aerial | Underwater | Object | **Mean** |
|---|---|---|---|---|---|---|---|---|---|---|---|
| ViT-S | ✔ | | 88.1 | 87.8 | 90.8 | 86.9 | 90.6 | 93.8 | 94.9 | 89.9 | 89.8 |
| | | ✔ | 92.9 | 93.0 | 98.4 | 94.4 | 95.7 | 96.4 | 99.2 | 96.6 | **95.3** |
| ViT-B | ✔ | | 91.2 | 91.9 | 95.7 | 90.2 | 90.9 | 96.4 | 94.9 | 96.6 | 92.9 |
| | | ✔ | 96.2 | 94.8 | 98.7 | 96.3 | 96.7 | 99.0 | 100 | 97.3 | **97.0** |
| ViT-L | ✔ | | 94.5 | 93.9 | 98.4 | 93.9 | 96.3 | 97.4 | 99.2 | 98.0 | 96.0 |
| | | ✔ | 96.4 | 93.9 | 99.0 | 96.3 | 97.3 | 99.5 | 99.2 | 98.0 | **97.1** |

Table 14: Per-scenario accuracy (%) of Depth Anything V2 on our proposed benchmark DA-2K.

### C.2   Comparison with the DIW dataset

Although DIW [11] and our DA-2K use the same annotation format (both sparse pixel pairs, we are inspired by DIW), our proposed DA-2K dataset is better in four aspects:

- **(more precise)** DIW is very noisy. For *most* pairs in DIW, we cannot decide the relative depth or hold the opposite opinion as the provided label. This can also be supported by MiDaS [7] that, better and larger models instead perform worse on DIW. In comparison, our DA-2K is precise, because we exclude many hard-to-decide or controversial pairs.

- **(better organized)** DIW randomly downloads images from Flickr, without carefully organizing. This would make users struggle to obtain straightforward insights from the evaluation results. In comparison, our DA-2K organizes all images by application scenarios, and thus can provide results for each individual application scenario.

- **(more diverse)** DIW images are typically collected from real life. However, considering the widespread application of MDE models in AIGC [101, 39], we provide additional non-real images, such as AI-generated images, cartoon images, *etc.*.
- **(high-resolution)** Most images in DIW have a low resolution of around 300×500, while we provide mostly 1500×2000 high-resolution images.

## C.3   Annotation details

To alleviate the burden of human annotators and avoid hard-to-decide pairs, we only pop out pixel pairs whose predicted depth ratio is larger than 3. For the evaluation scenarios of "transparent" and "object", we do not rely on model disagreement to pop out pairs. We simply manually analyze the images and select challenging pairs suited to the scenario. For other scenarios, we adopt both selection pipelines (*i.e.*, automatic disagreement-based selection and manual selection). In Table 15, we list the keywords we use to download images for each evaluation scenario.

| Evaluation scenario | Keywords |
| --- | --- |
| Indoor | room, home, living room, kitchen, bedroom, office, store, library, restaurant, museum, hall |
| Outdoor | road, outdoor, street, urban, rural, park, beach, mountain, downtown, alley, skyscraper, traffic, bridge, construction, parade, fireworks, festival, sporting event |
| Non-real (*e.g.*, AIGC, painting, *etc.*) | AI-generated, computer-generated, artwork, oil painting, impressionism, realism, abstract art, cartoon, animation, comic, caricature, illustration, fantasy, sci-fi, cyberpunk, alien, mythology |
| Transparent / reflective surfaces | glass, window, crystal, ice, water, transparent, clear, acrylic, plastic, reflective, mirror, see-through |
| Adverse style (*e.g.*, foggy, dark, *etc.*) | fog, dark, night, mid-night, overexposed, blur, snow, rain |
| Aerial | aerial, landscape, drone view, bird's eye view, city, cityscape, satellite view, top-down view |
| Underwater | underwater, ocean, sea, coral reef, diving, submarine, aquarium, marine life, shipwreck |
| Object | car, bicycle, motorcycle, airplane, bus, train, truck, boat, traffic light, fire hydrant, stop sign, parking meter, bench, bird, cat, dog, horse, sheep, cow, elephant, bear, zebra, giraffe, backpack, umbrella, sports ball, kite, baseball bat, cup, fork, knife, spoon, bowl, banana, apple, chair, bed, dining table, microwave, oven, toaster, sink, refrigerator, vase, scissors, teddy bear |

Table 15: Eight evaluation scenarios encompassed in our DA-2K. We use the keywords generated by GPT-4 to download images of corresponding scenarios on Flickr.

## C.4   Visualization

In Figure 19, we visualize some samples in our proposed DA-2K benchmark. They cover diverse representative scenarios and are of precise sparse annotations.

## D   Limitations

Currently, we use 62M unlabeled images for training. The computational burden is very heavy. Thus, in the future, we will study how to leverage such large-scale visual data more efficiently. Moreover, the current synthetic training sets are not diverse enough. We will attempt to collect synthetic images from more sources to train a more capable teacher model for better pseudo labeling.

Image                    Depth Anything V1                    Depth Anything V2

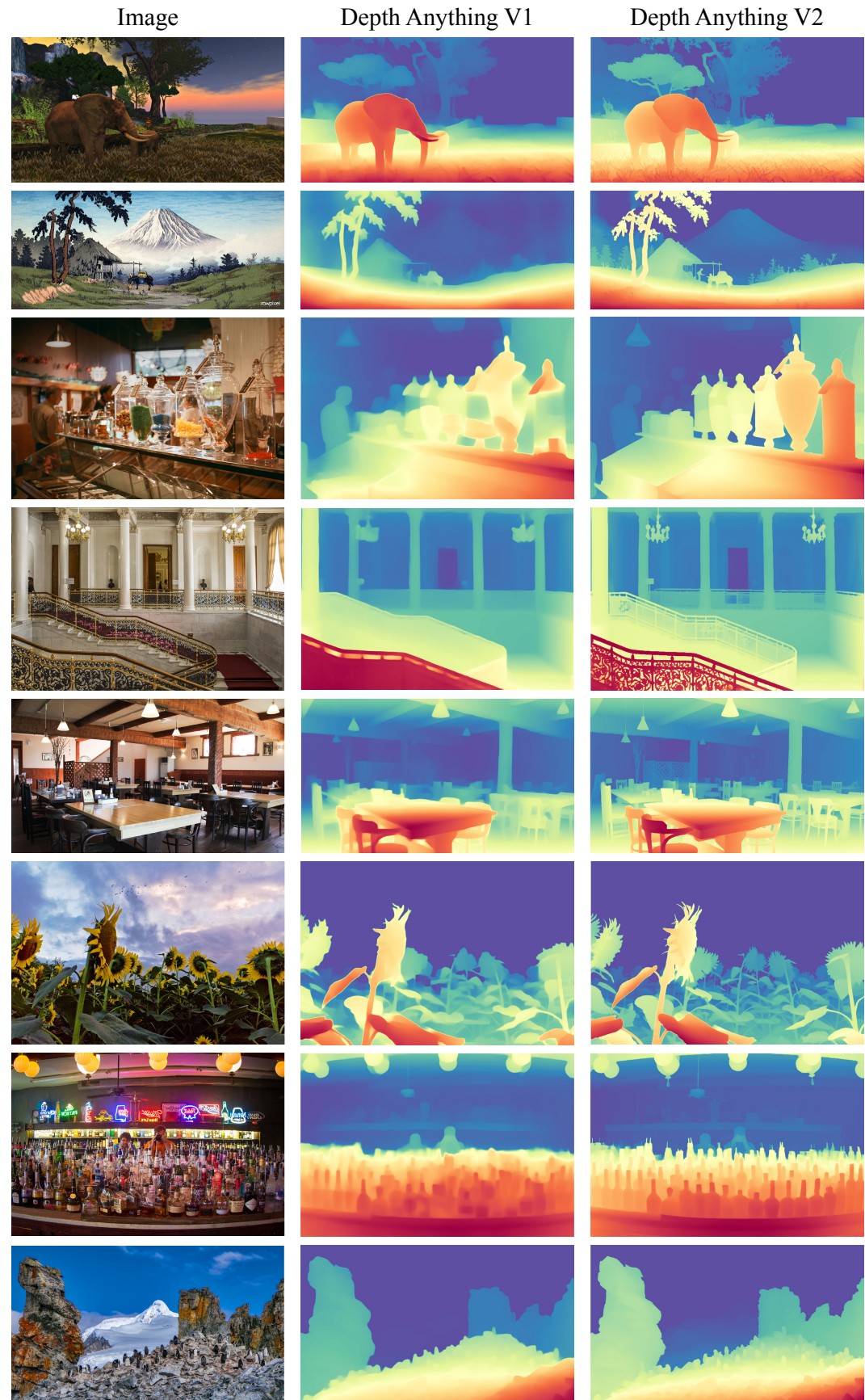

Figure 13: Comparison between Depth Anything V1 [89] and our V2 in open-world images.

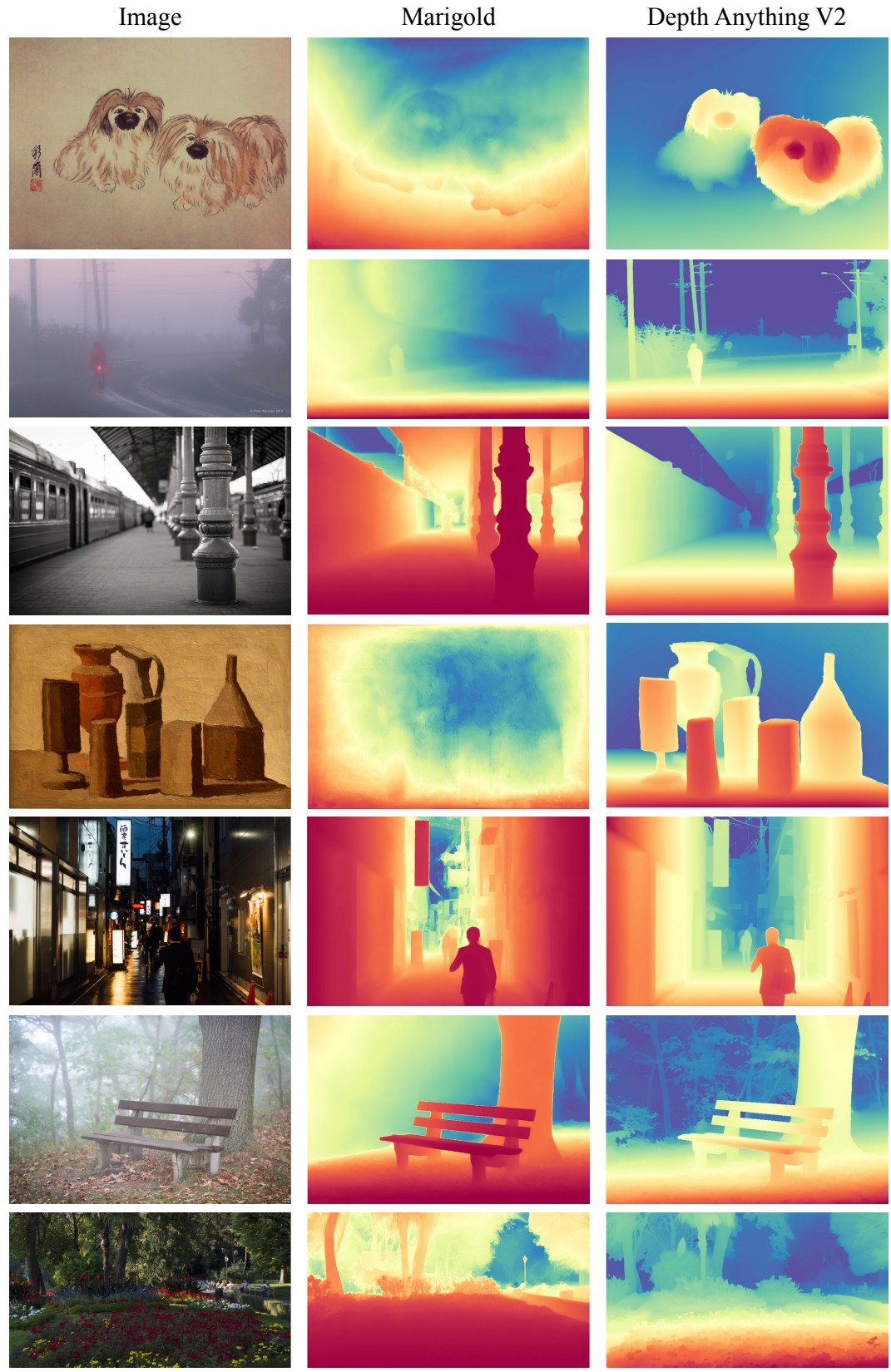

Figure 14: Comparison between Marigold [31] and our V2 in open-world images.

Image        ZoeDepth        Ours

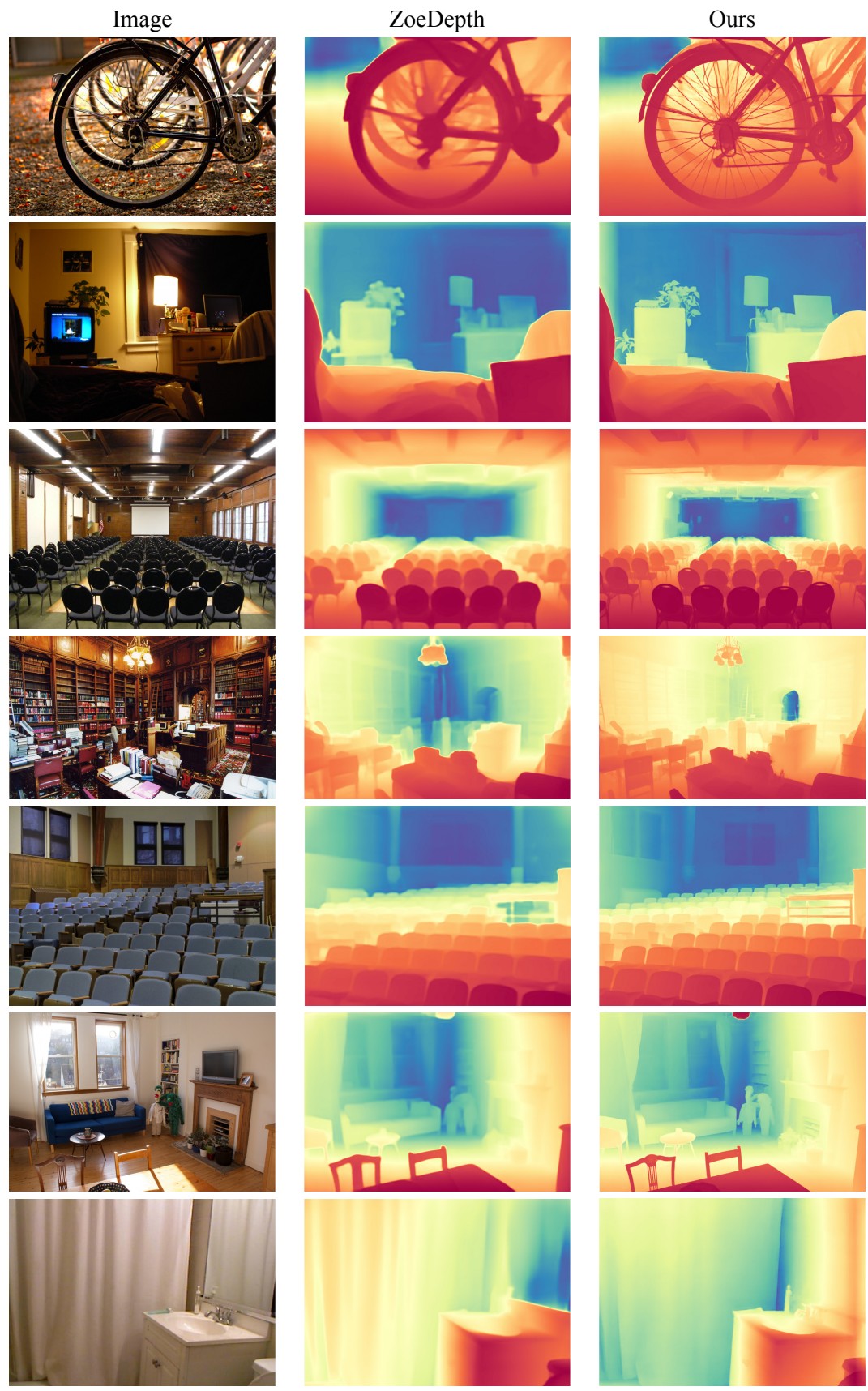

Figure 15: Comparison between ZoeDepth [6] and our fine-tuned metric depth model.

| Image | Labeled synthetic data | Pseudo-labeled real data |
|---|---|---|

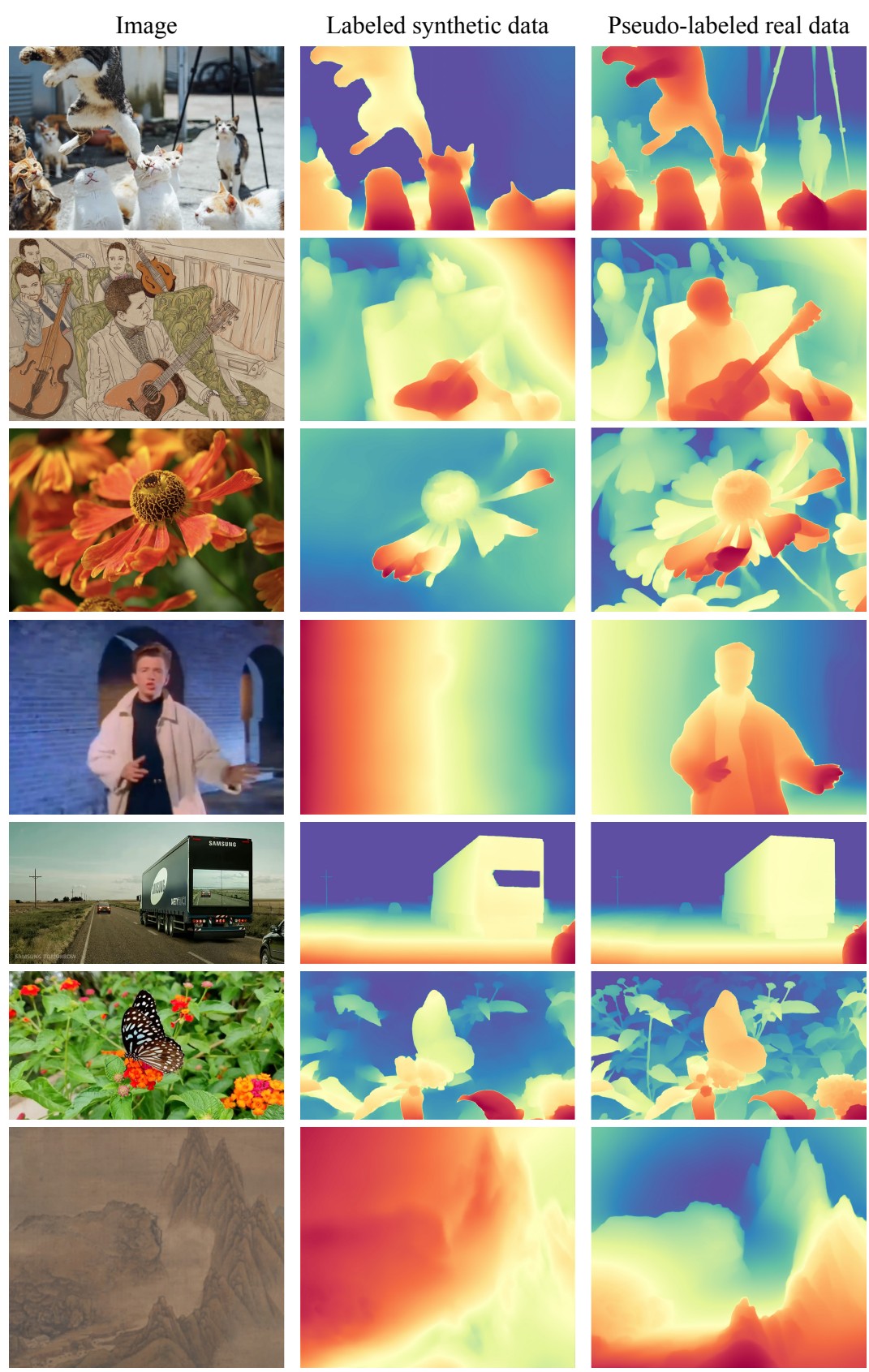

Figure 16: Qualitative comparison of the DINOv2-small-based depth model trained solely on labeled synthetic images and solely pseudo-labeled real images. The robustness is tremendously enhanced.

| Unlabeled image | Pseudo label | Unlabeled image | Pseudo label |

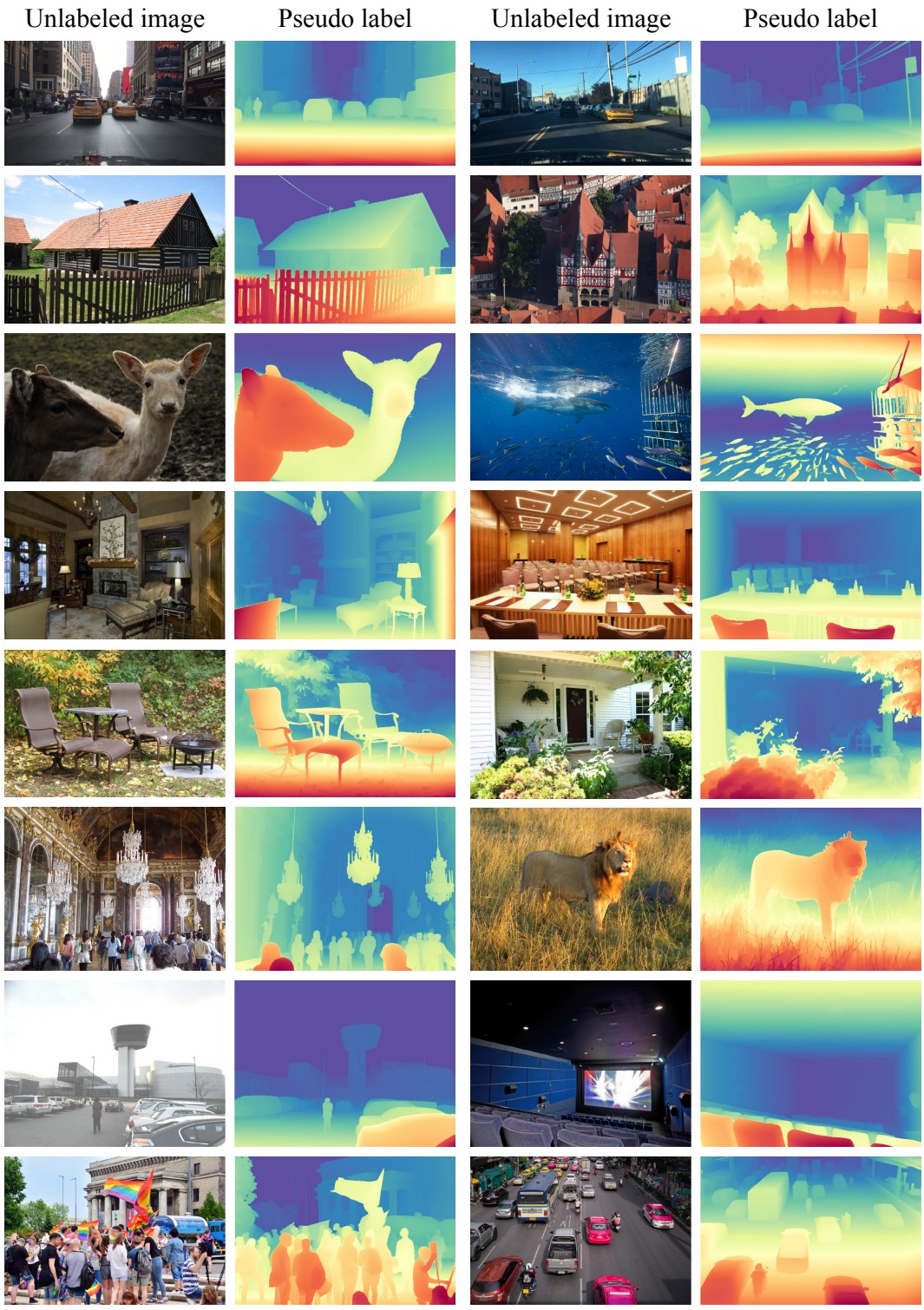

Figure 17: Visualization of our produced pseudo depth labels. From top to bottom, the highly diverse images are sampled from BDD100K [97], Google Landmarks [81], ImageNet-21K [60], LSUN [98], Objects365 [65], Open Images V7 [35], Places365 [103], and SA-1B [33] datasets, respectively.

Image      Depth Anything V1      Depth Anything V2

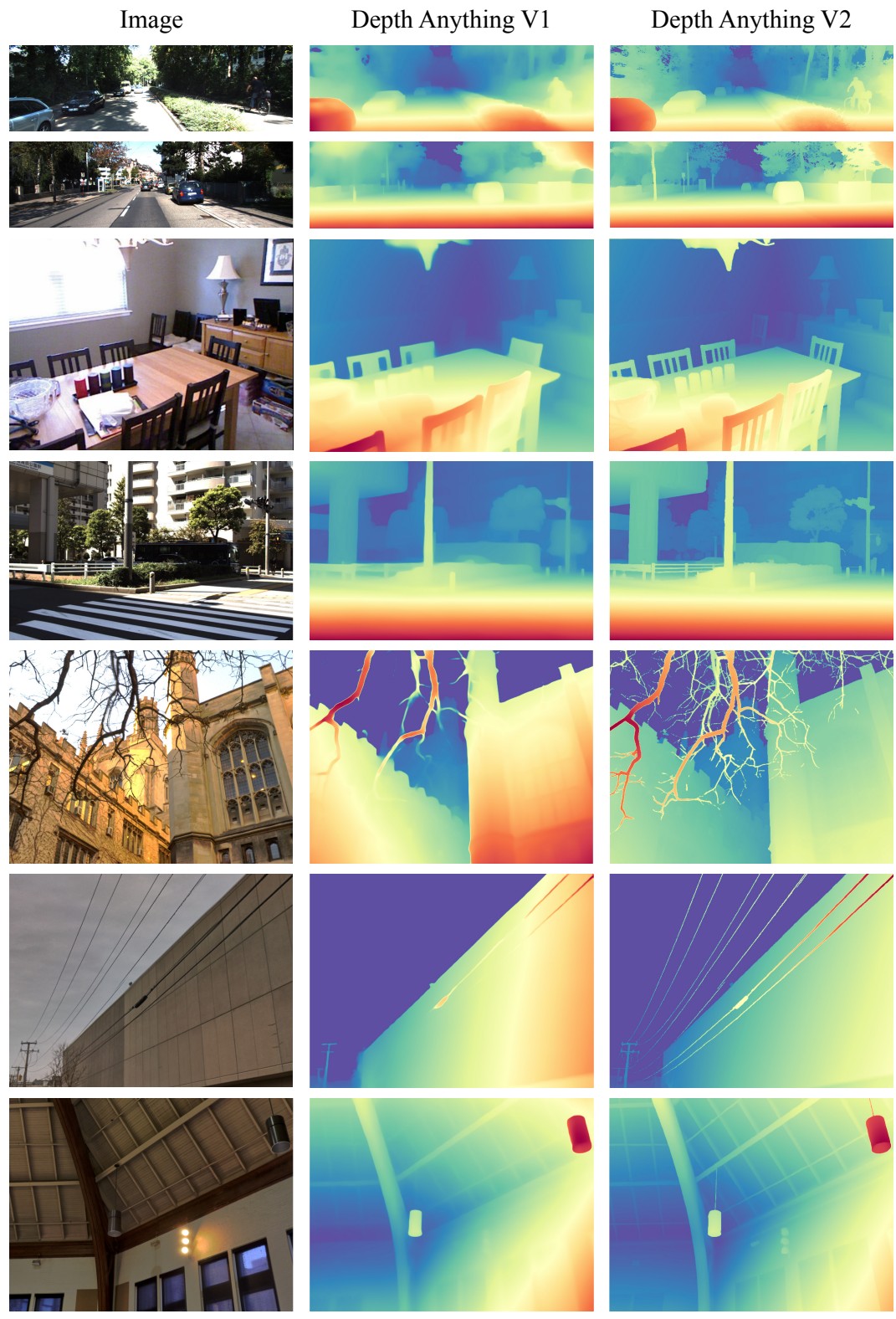

Figure 18: Qualitative results on widely adopted test benchmarks, *e.g.*, KITTI, NYU, and DIODE.

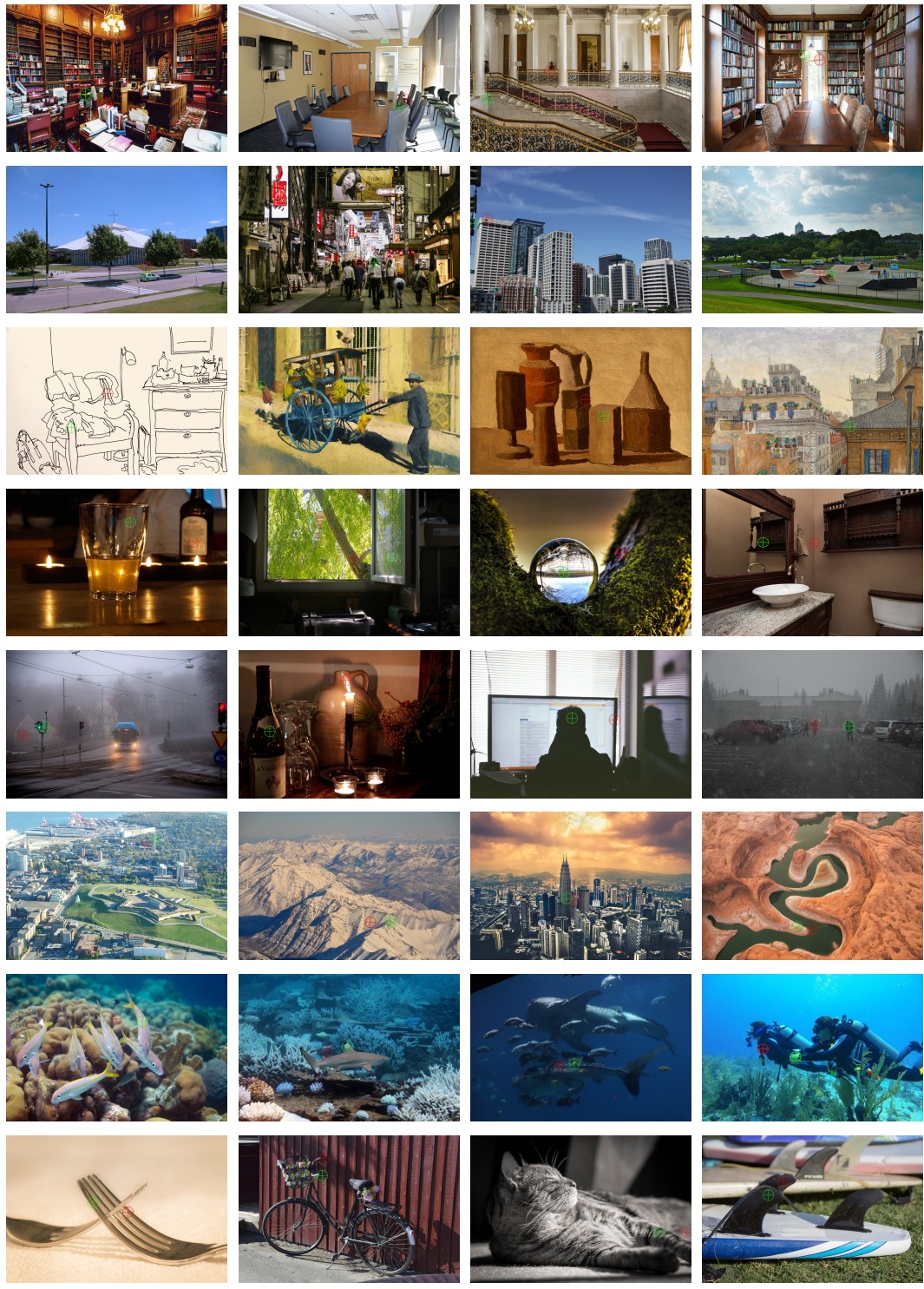

Figure 19: Visualization of images and precise sparse annotations on our benchmark DA-2K. Please **zoom in** to better view the annotated pairs. The green point is annotated as closer than the red point. From top to bottom, the images are sampled from indoor, outdoor, non-real, transparent/reflective, adverse style, aerial, underwater, and object scenarios, respectively.

