# OpenReview forum: "Depth Anything V2"
_NeurIPS.cc/2024/Conference — NeurIPS 2024 poster_

### Official Review · Reviewer_Bw2h · 2024-07-07

**Soundness:** 3
**Presentation:** 3
**Contribution:** 3
**Rating:** 5
**Confidence:** 4

**Summary:**

The paper tries to answer two important questions in current monocular depth estimation: 1. whether generative-based models can generate more fine-grained depth than discriminative models. 2. The effectiveness of synthetic data and large-scale pseudo data. The results show that discriminative models can also generate fine-grained depth estimation when trained with synthetic data and gradient matching loss. The paper provides several depth estimation models with different sizes, which may promote various downstream applications.

**Strengths:**

1. The paper proposes to enhance the performance of the existing discriminative depth estimation model, DepthAnything, with synthetic datasets and large-scale pseudo labels, which is simple yet effective.
2. The experiments are extensive and the writing is good.

**Weaknesses:**

1. How do you define the fine-grained depth quantitatively? There seem to be no concrete metrics to evaluate it, except qualitative visualization in the paper. So it is still hard to fairly compare the fine-grained depth with the generative-based models.
2. The pseudo label can only be relative depth, which may not be effectively directly used for metric depth estimation.
3. There are no dataset ablation experiments to verify the necessity of using the total 62M pseudo labels in the paper. If one random samples a subset from the pseudo-labeled real images, e.g. sample 10% data from all the 8 real datasets, what is the performance of the student model? Whether the 62M data is redundant or not?

**Questions:**

1. What does "+unlabeled real image" mean in Figure 6?
2. What is the backbone in Table 11?
3. In Section C.10, the authors find that synthetic data is important for fine-grained predictions, I am curious about the differences in quantitative results when training the teacher model with and without real labeled images.
4. Can 62M pseudo labels improve the performance of the teacher model (ViT-Giant backbone)?
5. For metric depth estimation, the authors fine-tuned the model from pre-trained relative depth estimation, what is the performance when fine-tuning the model from the original DINOv2 pre-trained backbone? (e.g. using the same ViT-large backbone with different weight initialization)
6. What is the performance of the proposed metric depth estimation model compared to recent SOTA models, e.g. Unidepth and Metric3Dv2?

**Limitations:**

see weakness and questions

---

> ### Author Rebuttal · Authors · 2024-08-07
>
> **Q1: Quantitative comparison on depth sharpness.**
>
> Quantitatively comparing different depth models in terms of sharpness is not trivial, mainly due to two reasons: 1) a lack of well-defined metrics for sharpness, and 2) the absence of *diverse, real, and fine-grained* benchmarks to evaluate this term. But to better address your concerns, we carefully select 140 annotated pairs that include thin objects or are near object boundaries from our diverse DA-2K benchmark. On these fine-grained challenging depth pairs, we achieve a superior accuracy of 98.6%, compared to Marigold's 88.6%.
>
> **Q2: Pseudo label of relative depth may not be effectively used for metric depth estimation.**
>
> In this work, we primarily focus on relative depth estimation, where we can fully unlock the strengths of large-scale unlabeled data. In comparison, obtaining pseudo metric depth for in-the-wild images without additional information is almost infeasible. Relative depth is also beneficial to metric depth. As shown in Table 4, a sufficiently pre-trained relative depth model can be quickly adapted to metric depth estimation with minimal fine-tuning (~1 hour on NYU-D and KITTI). Besides, pseudo labels of relative depth can serve as an auxiliary optimization target when jointly trained with the main metric depth estimation task. Using an auxiliary head to learn pseudo relative depth from large-scale unlabeled data can enhance the shared encoder's capabilities, benefiting the primary metric depth estimation task.
>
> **Q3: Whether the total 62M unlabeled images are redundant.**
>
> Thank you for your insightful question. Honestly, during the development of V2, we have also explored reducing the scale of 62M unlabeled images due to the almost unaffordable computational cost. We performed the following attempts:
>
> - Randomly selecting 50% unlabeled images from each unlabeled set (similar to the setting you mentioned).
> - Measuring the hardness of each unlabeled sample based on the agreement between the Depth Anything V1 and V2 predictions, *i.e.*, the higher the agreement, the easier the image. Based on the sorted hardness, we select 1) top 50% images, 2) medium 50% images, or 3) bottom 50% images, from each unlabeled set.
>
> Our comprehensive experiments reveal that all reduction strategies (1+3) result in performance drops of 0.5%-1% on all standard benchmarks and 1%-2% on our proposed DA-2K benchmark. Therefore, we conclude the full set of 62M unlabeled images is crucial for producing the most capable model.
>
> **Q4: Meaning of "+unlabeled real images" in Figure 6.**
>
> The main figure in Figure 6 illustrates the failure cases if the teacher model is purely trained on synthetic images. The upper-right subfigures labeled "+unlabeled real images" show the improved predictions of the model after it is re-trained on large-scale pseudo-labeled real images. We will clarify it in our final version.
>
> **Q5: The backbone used in Table 11.**
>
> The backbone used is ViT-Small. We will add this detail in our final version.
>
> **Q6: Training the teacher model with and without real labeled images.**
>
> In our early attempts, we have tried combining real labeled images (e.g., Cityscapes, DIML) to train our teacher model. They will bring a 0.3% improvement on the NYU-D and KITTI datasets. However, we observed a significant degradation in depth sharpness and robustness to transparent/reflective surfaces due to flawed GT labels in the real datasets. Replacing these low-quality labels with our re-annotated pseudo labels provides a similar improvement (~0.5%) when the re-annotated images closely match the test domains (*e.g.*, DIML for NYU-D). However, To maintain model universality and avoid overfitting to specific test metrics, we opted to use only the eight most representative large-scale unlabeled datasets.
>
> **Q7: Whether the 62M pseudo labels can improve ViT-Giant-based teacher model.**
>
> For standard benchmarks' metrics (*e.g.*, NYU-D, KITTI), we find the 62M pseudo-labeled images are not necessary to the ViT-Giant-based teacher model. As we have repetitively mentioned in the paper, this is mainly due to the label noise in these existing test benchmarks (Figure 8). In comparison, on our proposed high-quality DA-2K benchmark, these additional images can further improve the teacher's performance by 1%. Moreover, these pseudo-labeled real images enhance the model's robustness in challenging cases, as demonstrated in Figure 6 (*i.e.*, sky and dark person).
>
> **Q8: The performance of fine-tuning original DINOv2 weights for metric depth estimation.**
>
> This experiment is ablated in Depth Anything V1. Fine-tuning DINOv2 on NYU-D and KITTI achieves $\delta_1$ scores of 0.973 and 0.971, respectively. In contrast, our fine-tuned Depth Anything V2 obtains 0.984 and 0.983 results on the two datasets, demonstrating significantly better performance than the original DINOv2.
>
> **Q9: Comparison with metric depth models UniDepth and Metric3Dv2.**
>
> Thank you for pointing out these two competitive methods. We did not compare with UniDepth and Metric3Dv2 mainly because they were released within two months before our submission. According to the [NeurIPS policy](https://neurips.cc/Conferences/2024/PaperInformation/NeurIPS-FAQ), we are considered concurrent works and are not expected to compare with them. Briefly compared here: on NYU-D, our fine-tuned metric depth model achieves the same $\delta_1$ result as UniDepth and is 0.5% lower than Metric3Dv2. It is worth noting that we require significantly fewer labeled images than them (ours: 0.6M *vs.* theirs: 16M and 3M). Their training sets already cover many similar scenes (*e.g.*, Taskonomy, ScanNet) as NYUv2, leading to better transfer results. More importantly, our focus is *relative* depth rather than *metric* depth. Our relative depth model performs much more robustly, as shown in the PDF. Quantitatively, on our proposed DA-2K benchmark containing mostly in-the-wild images, we achieve over 10% higher accuracy than them.

---

> > ### Comment · Reviewer_Bw2h · 2024-08-11
> > **Thanks for your feedback**
> >
> > Thank you for the feedback. Similar to Reviewer RjZq, could you provide a more detailed evaluation comparing the relative depth metrics of existing metric depth models, such as Metric3D, Metric3D v2, and Uni-Depth, against challenging indoor and outdoor benchmarks? Specifically, NYU-D is known for having noisy labels, which raises a concern. It appears that metric depth models trained on large real-world datasets may already exhibit strong generalization performance across diverse scenes, suggesting that pseudo-labeling might not be essential. Consequently, the main benefit of the pseudo-labeling approach proposed in this paper seems to be in achieving more detailed relative depth maps and cannot be applied to improve metric depth estimation directly since pseudo-labels lack scale information.

---

> > > ### Author Response · Authors · 2024-08-11
> > > **Thank you for further feedback**
> > >
> > > To verify the relative-depth generalization ability of existing metric depth models, we have evaluated UniDepth and Metric3D (v1 & v2) on our proposed challenging and precise DA-2K benchmark, which encompasses eight diverse scenarios (*e.g.*, indoor, outdoor, adverse style, see Figure 9(b) for details). Results are summarized below.
> > >
> > > |  Method   | Indoor | Outdoor | Average of 8 scenarios |
> > > | :----  | :----:  | :----:  | :----:  |
> > > |  UniDepth | 80.4 | 84.2 | 83.1 |
> > > | Metric3Dv1 | 83.5 | 86.8 | 83.7 |
> > > | Metric3Dv2 | 85.7 | 88.1 | 86.2 |
> > > | **Depth Anything V2 (Ours)**| **89.8** | **95.1** | **94.8** |
> > >
> > > Moreover, in Table D.1 of our appendix, we have demonstrated the indispensable role of large-scale unlabeled images. Part of the results are borrowed here:
> > > |  Using unlabeled data?   | Indoor | Outdoor | Average of 8 scenarios |
> > > | :----  | :----:  | :----:  | :----:  |
> > > |  No | 85.3 | 92.9 | 91.4 |
> > > | **Yes** | **89.8 (+4.5)** | **95.1 (+2.2)** | **94.8 (+3.4)** |

---

> > > > ### Comment · Reviewer_Bw2h · 2024-08-12
> > > > **Thanks for authors feedback**
> > > >
> > > > The results indicate that DepthAnythingV2 outperforms metric depth models on relative metrics. I have an additional question: could you provide a more detailed explanation of how the gradient matching loss is implemented in disparity space?

---

> ### Author Response · Authors · 2024-08-12
> **Thanks for further response and the implementation of gradient matching loss**
>
> Thank you for acknowledging our better performance on the relative depth metrics.
>
> Regarding the gradient matching loss, it was originally proposed by MiDaS. We have provided its formulations in the one-page PDF.  And our implementation is adopted from MiDaS, which can be found here: https://gist.github.com/dvdhfnr/732c26b61a0e63a0abc8a5d769dbebd0#file-midas_loss-py-L93-L113. The groundtruth label in this gradient matching function is in the disparity space, via inversing the depth label (*i.e.*, 1 / depth) and re-normalizing it to 0-1 with min-max value.

---

> ### Author Response · Authors · 2024-08-13
>
> Dear Reviewer Bw2h, as the author-reviewer rebuttal deadline approaches, we want to follow up on our recent response to your additional question regarding our gradient matching loss. We’ve provided the [source code](https://gist.github.com/dvdhfnr/732c26b61a0e63a0abc8a5d769dbebd0#file-midas_loss-py-L93-L113) and explanations above for the detailed implementation of this loss.
>
> If you have any further comments or need additional clarification, we would be more than happy to assist. If our response has satisfactorily addressed your concerns, we kindly ask if you might consider revisiting your initial score.
>
> Thank you for your precious time and consideration.

---

### Official Review · Reviewer_fDqs · 2024-07-09

**Soundness:** 4
**Presentation:** 4
**Contribution:** 3
**Rating:** 7
**Confidence:** 4

**Summary:**

Depth Anything V2 is an advanced monocular depth estimation model that improves upon its predecessor by utilizing synthetic images, enhancing the teacher model's capacity, and leveraging large-scale pseudo-labeled real images for training. This approach results in significantly faster and more accurate depth predictions compared to models based on Stable Diffusion. This model also offers a range of model sizes to support various applications and demonstrates strong generalization capabilities, which are further fine-tuned with metric depth labels. Additionally, a versatile evaluation benchmark, DA-2K, has been developed to address the limited diversity and noise in current test sets, facilitating more accurate future research.

**Strengths:**

The paper offers several noteworthy advantages. Firstly, it provides a comprehensive analysis of recent depth foundation models, detailing their features, strengths, and weaknesses. The comparison between generative and discriminative models in Table 1 is particularly insightful. Secondly, the approach is novel, utilizing pseudo-labeled real images to harness the advantages of both synthetic and real datasets. This method is impressive and appears beneficial for scaling up foundation models, similar to SAM. Thirdly, the paper demonstrates strong generalization performance, as evidenced by various tables and figures. The real-time performance showcased in the CVPR conference demo was particularly impressive. Lastly, the extensive additional experiments and analyses in the appendix offer valuable insights, supporting the validity of the proposed methods and experimental protocols.

**Weaknesses:**

Data Volume and Performance Relationship: According to Table 10, there seems to be a linear relationship between the amount of data and performance. It would be interesting to see how the model performs with even more data added.

Metric Depth Estimation Comparison: Table 4 shows that the results of Metric Depth Estimation are similar to or worse than other Monocular Depth Estimation models (UniDepth[1], Metric 3D[2]). Additional analysis on this would be beneficial.


##### [1] UniDepth: Universal Monocular Metric Depth Estimation
##### [2] Metric3D v2: A Versatile Monocular Geometric Foundation Model for Zero-shot Metric Depth and Surface Normal Estimation

**Questions:**

These points are included in the Weakness.

**Limitations:**

The limitations of Depth Anything V2 include a dependency on data volume for performance, as suggested by Table 10, which raises questions about scalability and diminishing returns with more data. Additionally, Table 4 shows that its Metric Depth Estimation results are comparable to or worse than other models like UniDepth and Metric 3D v2, indicating that while the model excels in some areas, it does not consistently outperform competitors across all metrics.

---

> ### Author Rebuttal · Authors · 2024-08-07
>
> **Q1: The performance if even more data is added.**
>
> Thank you for raising this insightful question. Currently, we have used 62M unlabeled images from eight *highly curated* public datasets, *e.g.*, SA-1B and Open Images. As you mentioned, based on the positive scaling curve of experiments in Table 10, we believe that if more *curated* data is introduced, the model's generalization ability will be further enhanced. Unfortunately, as far as we know, **there are no ***curated*** public datasets available that can further ***significantly*** scale up the existing 62M images (***e.g.***, scaling up to 100M)**. Some web-scale datasets, like LAION (no longer public due to privacy concerns) and DataComp, exhibit poorer image quality and imbalanced visual concepts, which have been shown to be even detrimental to model learning [1]. Therefore, at this stage, it is challenging to further scale up our unlabeled pool from existing public sources. We will leave this interesting topic for future work. We may first curate higher-quality raw images from web-scale billion-level datasets similar to [1], and then pseudo-label them with precise depth annotations for our models to learn.
>
> On the other hand, although we cannot further *significantly* scale up the unlabeled data, **it is feasible to involve a certain amount of unlabeled images from ***specific domains*** to enhance model performance on these targeted domains**. For example, we have tried adding the NYU-D or KITTI *raw unlabeled* training images (~20K) to our large-scale unlabeled pool. We found that the test results on the two datasets are both boosted by nearly 0.5% (the baseline results are already very competitive). These results further demonstrate the importance of narrowing the domain shift between training and test data. This observation aligns with our motivation in the paper to incorporate unlabeled real images to mitigate the disadvantages of synthetic images (*i.e.*, distribution shift and poor diversity) while amplifying their strengths (*i.e.*, high precision).
>
> [1] Automatic Data Curation for Self-Supervised Learning: A Clustering-Based Approach, arXiv 2024.
>
>
> **Q2: Comparison with UniDepth and Metric3Dv2.**
>
> Thank you for pointing out these two competitive methods. We did not compare with UniDepth and Metric3Dv2 mainly because they were released within two months before our submission. According to the [NeurIPS policy](https://neurips.cc/Conferences/2024/PaperInformation/NeurIPS-FAQ), we are considered concurrent works and are not expected to compare with them. Briefly compared here: on NYU-D, our fine-tuned metric depth model achieves the same $\delta_1$ result as UniDepth and is 0.5% lower than Metric3Dv2. It is worth noting that we require significantly fewer labeled images than them (ours: 0.6M *vs.* theirs: 16M and 3M). Their training sets already cover many similar scenes (*e.g.*, Taskonomy, ScanNet) as NYUv2, leading to better transfer results. More importantly, our focus is *relative* depth rather than *metric* depth. Our relative depth model performs much more robustly, as shown in the PDF. Quantitatively, on our proposed DA-2K benchmark containing mostly in-the-wild images, we achieve over 10% higher accuracy than them.

---

> > ### Comment · Reviewer_fDqs · 2024-08-12
> >
> > My questions have been fully addressed, and I am satisfied with the response. I would like to express my gratitude to the authors for their thorough and thoughtful answers. I will maintain the decision to accept.

---

### Official Review · Reviewer_RjZq · 2024-07-11

**Soundness:** 3
**Presentation:** 3
**Contribution:** 2
**Rating:** 6
**Confidence:** 5

**Summary:**

The authors introduce Depth Anything V2, a powerful monocular depth estimation model. This model relies entirely on synthetic data to train a teacher depth estimation model, which is then used to generate pseudo-labeled real images. These pseudo-labeled images are subsequently fed into the training pipeline. Depth Anything V2 achieves state-of-the-art results in both quantitative and qualitative evaluations. Additionally, the authors offer models in various scales, accommodating different scenarios. To address the limitations of existing monocular depth estimation benchmarks, the authors also provide a new benchmark with precise annotations and diverse scenes, facilitating future research. However, compared to depth anythingv1 and other affine-invariant depth methods, the novelty is limited.

**Strengths:**

1.	The authors dedicate considerable space to discussing the choice of datasets, highlighting the disadvantages of labeled data, the advantages and limitations of synthetic images, and the role of large-scale unlabeled real images. This provides a clear rationale for the proposed pipeline.
2.	A new benchmark is introduced to address the limitations of existing benchmarks. This benchmark encompasses a wide variety of scenarios and includes comprehensive, high-resolution depth estimation results.
3.	The results are impressive. The method demonstrates significant improvements in zero-shot relative depth estimation, particularly on the proposed DA-2K benchmark. The qualitative results also reveal the fine-grained detail of the predictions.

**Weaknesses:**

1. Lack of Novelty: The article primarily focuses on analyzing the choice between synthetic and real data, while the pipeline remains unchanged compared to the original Depth Anything V1. V2 used more high-quality data to achieve better performance, but it has no new methods contributions and insights.

2.	Lack of Detail: Although some methods may be well-known within the community, it would be beneficial for the authors to elaborate more on the description of the pipeline and the formulation of the loss function.

3.	More Comparisons Required: In Table 2, the authors argue that the method predicts affine-invariant inverse depth and compares it with Depth Anything V1 and MiDaS V3.1. MiDaS is not an up-to-date SOTA method. It should enclose more recent works, such as UniDepth, ZoeDepth, Marigold, Metric3D, and Metric3Dv2. Similarly, in Table 4, which compares metric depth methods, Unidepth, and metric3dv2 achieve state-of-the-art performance in these two benchmarks, but they are not listed in tables.

**Questions:**

1.	Why can the proposed method be compared with other methods on the proposed dataset, but not on other zero-shot relative depth estimation datasets?
2.	The performance of the pipeline seems to be limited by the performance of the teacher model, as the training data for the student models is provided by the teacher model. Why, then, do some student models outperform the teacher model in Table 5?

**Limitations:**

The paper does not present limitations in detail.

---

> ### Author Rebuttal · Authors · 2024-08-07
>
> **Q1: Our contributions and novel insights.**
>
> We do not position the pipeline as our contribution. There are many dimensions to measuring a paper's contributions. From the very start of our paper (L1-3, L41-45), we emphasize that, *instead of proposing a new module or pipeline, this work is centered on the **data perspective** and aims to reveal crucial insights:*
>
> - **[Insight on flawed training data]** Efficient discriminative models can also produce fine-grained depth predictions. Heavy diffusion models are not necessary. *Replacing flawed real images with precise synthetic images* is the key to the prediction sharpness. There is NO prior work that achieves both fine-grained predictions and ultra-efficient inference (>10x faster than SD-based models). So we believe this insight is critical and worth presenting to our community.
> - **[Insight on scaling up teacher model and the *new* role of unlabeled real images]** We reveal that, however, *it is non-trivial to leverage synthetic images* (Figure 5, 6, Table 13). To overcome the disadvantages (distribution shift and poor diversity) of synthetic images and amplify their strengths (preciseness), we present a simple solution: use the most capable DINOv2-Giant-based teacher model to learn from precise synthetic images first, and then pseudo-label diverse real images for students to learn from. *This pipeline is well-motivated by the carefully studied properties of synthetic images.*
> - **[Insight on flawed evaluation benchmarks & our contribution: DA-2K]** We point out the drawbacks of current test benchmarks: noisy annotation, poor diversity, and low resolution. In response, we construct a versatile benchmark DA-2K, with precise annotations, rich diversity, and high-resolution images. DA-2K is a valuable supplement to existing benchmarks.
>
> All these novel insights combined contribute to a more capable Depth Anything V2 for our community. *Our data-centric contribution is acknowledged by Reviewer rTF8 and fDqs.* We politely hope you can reconsider our work's contributions and insights from the data perspective. Thank you.
>
> **Q2: More details on the pipeline and loss function.**
>
> Thank you for your kind reminder. We will detail them in our final version. For now, we provide a preview in the PDF.
>
> **Q3: Comparison with more methods on relative and metric depth estimation.**
>
> Regarding relative depth, we only compared with MiDaS and Depth Anything V1 because we all predict the *inverse* depth (we prefer inverse depth due to its [numerical stability](https://github.com/isl-org/MiDaS/issues/21)), while Marigold produces the depth *without inversion*. It will cause some evaluation noise when converting the two depths to the same space, mainly arising from the [clipping practice](https://github.com/prs-eth/Marigold/blob/f74115261b67b59fb536994d0413f64d69af65c5/eval.py#L198-L200). However, to address your concerns, we further compare our method with Marigold. Our method achieves much higher $\delta_1$ scores than Marigold on KITTI (94.4% *vs.* 91.6%) and NYU-D (98.0% *vs.* 96.4%).
>
> Regarding metric depth, we have initially compared with your mentioned ZoeDepth (Table 4). We did not compare with UniDepth and Metric3Dv2 mainly because they were released within two months before our submission. According to the [NeurIPS policy](https://neurips.cc/Conferences/2024/PaperInformation/NeurIPS-FAQ), we are considered concurrent works and are not expected to compare with them. Briefly compared here: on NYU-D, our fine-tuned metric depth model achieves the same $\delta_1$ result as UniDepth and is 0.5% lower than Metric3Dv2. It is worth noting that we require significantly fewer labeled images than them (ours: 0.6M *vs.* theirs: 16M and 3M). Their training sets already cover many similar scenes (*e.g.*, Taskonomy, ScanNet) as NYUv2, leading to better transfer results. More importantly, our focus is *relative* depth rather than *metric* depth. Our relative depth model performs much more robustly, as shown in the PDF. Quantitatively, on our proposed DA-2K benchmark containing mostly in-the-wild images, we achieve over 10% higher accuracy than them.
>
> **Q4: Why our method is compared with other methods on our proposed DA-2K benchmark (Table 3), but not on other benchmarks (Table 2).**
>
> We explained in Q3 that we do not compare with Marigold in Table 3 due to concerns on the depth inversion noise during evaluation (see Q3 for results). However, we can compare with Marigold on our proposed DA-2K benchmark because DA-2K is a sparsely annotated test set. It only requires the model to discriminate which point between a pair is closer. Therefore, any depth model can be easily compared without inversion noise.
>
> **Q5: Why some student models can outperform their teacher model.**
>
> This is indeed a very common phenomenon in semi-supervised learning \[1, 2, 3\], where a typical strategy called "self-training" pseudo-labels unlabeled data to expand the training set. Limited by space, an intuitive explanation for this effectiveness is that explicitly training on extra pseudo-labeled data better shapes the model's decision boundary for more robust generalization. A similar phenomenon is observed in Depth Anything V1 (its Table 9, row 1 *vs.* row 4). Additionally, Llama 3.1 produces synthetic data for itself to iteratively bootstrap its performance. An extreme but intuitive example from [Hinton's recent talk](https://www.youtube.com/watch?v=n4IQOBka8bc&t=843s) also illustrates this: *a classifier trained on 50% noisy labels can achieve much higher than 50% test accuracy*. As Hinton states, "It can 'see' the training data is wrong. *Students can be smarter than the advisor*".
>
> [1] Pseudo-Label: The Simple and Efficient Semi-Supervised Learning Method for Deep Neural Networks, ICML Workshop 2013.
>
> [2] Self-training with Noisy Student improves ImageNet classification, CVPR 2020.
>
> [3] FixMatch: Simplifying Semi-Supervised Learning with Consistency and Confidence, NeurIPS 2020.

---

> ### Comment · Reviewer_RjZq · 2024-08-10
> **Comment**
>
> Thanks for the authors' detailed reply. Most of my concerns have been solved.
>
> However, in Q3, I sill believe the papaer should includ more methods for comparisons.
> Firstly, affine-invariant (or relative depth present in your answer) is defined as the scale-shift invariant to the metric depth. All downstream applications are in depth space, such as recovering the 3D point cloud. Although some methods train their model in the inverse space, the result should be in the depth space instead of inverse depth space. Thus in comparisons, the paper should not exclude methods, such as marigold, geowizard, leres, and others, that output the affine-invariant depth, but only choose MiDaS. There are more advanced methods.
> Furthermore, recent metric depth methods, such as zoedepth, zerodepth, metric3D also present affine-invariant depth comparisons. I believe this comparison decouples the metric and can better present the geometry quality of the depth. The paper can consider include them for comparison. Affine-invariant (or relative depth) is only a compromise representation, as the metric depth prediction is very ill-posed.

---

> > ### Author Response · Authors · 2024-08-12
> > **Thank you for acknowledging our feedback and raising the score**
> >
> > Thank you for acknowledging our further feedback and raising the score to 6. We will include your constructive feedback into our final version. Thank you!

---

> ### Author Response · Authors · 2024-08-10
> **Thank you for futher feedback**
>
> Thank you for your kind suggestions regarding the methods we compared. We agree with your recommendations and will ensure that all mentioned methods, including both affine-invariant and metric depth estimation, are included in our final verison. The only exception is ZoeDepth, since it is trained exclusively on NYU-D and KITTI, which cannot be considered *zero-shot* depth estimation (to NYU-D or KITTI) as ours and other methods you mentioned. But our fine-tuned metric depth models significantly outperform ZoeDepth in metric depth estimation.
>
> Briefly comparing our method with other methods (some results are borrowed from Metric3D): on NYU-D: DiverseDepth achieves the $\delta_1$ score of 87.5%, LeReS is 91.6%, HDN is 94.8%, ZeroDepth is 92.6%, Metric3D is 96.6%, Marigold is 96.4%, Geowizard is 96.6%, while ours is **98.0%**. We acknowledge that there are differences in the training settings among these methods, particularly in terms of the coverage of training data and model architecture. In the final version, we will also make an effort to highlight these differences to better showcase the strengths of each method from various perspectives. Thank you again for your valuable advice.

---

### Official Review · Reviewer_rTF8 · 2024-07-15

**Soundness:** 4
**Presentation:** 4
**Contribution:** 4
**Rating:** 7
**Confidence:** 5

**Summary:**

A system is proposed for scaling up large monodepth estimation models to achieve very strong zero-shot performance, focusing on finely detailed depth prediction. Essentially, a teacher model is trained on synthetic depth datasets with perfect ground truth, and thereafter distilled on a large in-the-wild dataset.

**Strengths:**

* The qualitative results are really stunning.
* The distillation-based framework is novel (as far as I know), and provides a valuable insight into the training of depth estimators more generally.

**Weaknesses:**

* It would be ideal to show the results back-projected into 3D as a point cloud or mesh for some example cases. It is difficult to tell the depth map quality from colormaps alone.
* The comparison with MIDAS feels like it leaves out an important limitation of the proposed the method. The proposed high-quality synthetic datasets appear to consist mostly of indoor and outdoor scenes. How will the technique stack up against MIDAS for unusual images that are not likely to be well-represented in such synthetic data, but may be available in 3D movies, such as of moving animals or people? Does the distillation on in-the-wild images just solve this problem?
* The paper appears to sort of conflate image-conditioned generation and regression, which are different problems even for image-conditional depth estimation. While "blurry" depth maps may be suboptimal for users looking for perceptual quality and seeking to back-project a 3D pointcloud or mesh, they may be in some sense metrically optimal since even scale and shift-invariant depth estimation is often ambiguous. Making predictions "fine-grained" could just be a problem of learning to sample from the distribution of image-conditioned depth, rather than heuristically applying losses such as gradient-based matching losses to a regression-based framework which is inherently a flawed way to look at the problem of perceptually high quality depth estimation.

**Questions:**

The performance on scenes which are not among the original synthetic data (animals, people, etc.) is unusually impressive. I understand that distillation is the proposed explanation, but can the authors provide a clear intuition or explanation for why this works so well?

**Limitations:**

The computational limitations are briefly mentioned.

---

> ### Author Rebuttal · Authors · 2024-08-07
>
> **Q1: Back-project the depth map into 3D as a point cloud.**
>
> Thank you for your valuable advice. We have added some visualizations of the back-projected point clouds in the one-page PDF. Due to space constraints, we were only able to include a limited number of examples. However, we will try to include more comprehensive visualizations in the final version to provide a clearer demonstration of the effectiveness of our method.
>
>
> **Q2: Why our method is impressive even on unusual images/objects (***e.g.***, moving animals) that are not well-represented in our synthetic training sets.**
>
> Thank you for raising this insightful question. We provide more comparisons with MiDaS on moving animals and people in the one-page PDF. Our strong generalization ability can be attributed to three key factors working together:
>
> - **[Object commonality]:** Even if some real objects are absent from our synthetic sets, there exist synthetic objects that share features with these real objects. For instance, synthetic moving cars in our Virtual KITTI dataset exhibit dynamics similar to real moving animals. Synthetic tables in our Hypersim dataset provide general object structure knowledge (*e.g.*, four legs and one body) for our model to generalize to animals (*e.g.*, elephants) with similar structures (the occlusion relationship among legs is similar). Such widespread commonality equips our model with a basic and general understanding of the physical world.
> - **[Strong pre-trained encoder]:** While object commonality helps, most pre-trained models still struggle to generalize robustly to novel new objects. As shown in Figure 5, vision encoders BEiT-Large, SAM-Large, and SynCLR-Large all fail to produce satisfactory results on the simple cat image. In contrast, the DINOv2-Giant encoder succeeds in such synthetic-to-real transfer. DINOv2-Giant, with over 1B parameters and pre-trained on 142M curated data, possesses rich prior knowledge about the real world, allowing it to generalize well even only fine-tuned with our 595K synthetic images.
> - **[Distillation from diverse pseudo-labeled real images]:** While the first two points ensure reliable generalization to real objects in many cases, they are not perfect (Figure 6). Distillation from diverse pseudo-labeled real images provides the final assurance. Our model harnesses the most general and accurate representations from the predominantly correct pseudo labels, overcoming the noise within them. An extreme but intuitive example from [Hinton's recent talk](https://www.youtube.com/watch?v=n4IQOBka8bc&t=843s) illustrates this: *a classifier trained on 50% noisy labels can achieve much higher than 50% test accuracy*. As Hinton states, "It can 'see' the training data is wrong. *Students can be smarter than the advisor*". Additionally, we discard the largest-loss (top 10%) regions during re-training (L162), ensuring our model learns from cleaner pseudo labels. Qualitatively, Figure 15 shows the tremendous advantages of using pseudo-labeled real data for training.
>
>
> **Q3:** **Whether our "fine-grained" depth maps are merely *perceptually* high-quality, but are indeed *metrically* worse than "blurry" depth maps.**
>
> **Our depth maps are both *perceptually* high-quality and *metrically* comparable or better than SOTAs.** In Table 2, we measure the quantitative results of depth models by fitting a scale and a shift scalar to align the predicted affine-invariant depth to GT metric depth. Our results are much better than MiDaS and comparable or better than Depth Anything V1. For the partially comparable results, we analyze (L171-174) that current test sets are too noisy and coarse to exhibit the true strengths of our models (*e.g.*, fine-grained predictions, robust to reflective and transparent surfaces). Motivated by the limitations of existing evaluation benchmarks, we build a new benchmark DA-2K, with diverse scenes, precise annotations, and high-resolution images (Section 6). Our V2 achieves more than 10% higher accuracy than V1 on our proposed high-quality benchmark, as shown in Table 3.
>
> **Gradient matching loss $L_{gm}$ is indeed also beneficial to optimizing scale-shift invariant loss.** Initially, sharing similar concerns with yours, we attempted to remove $L_{gm}$. Unfortunately, we observed all metrics in Table 2 degraded (more than 1% drop in the $\delta_1$ metric). For this phenomenon, we argue that *$L_{gm}$ is also beneficial in optimizing the affine-invariant depth to be globally optimal*, because a perfectly predicted depth should achieve a zero $L_{gm}$ value compared with GT depth map. For example, the GT depth of synthetic images is not only *perceptually* high-quality, but also *metrically* precise in the back-projected 3D space, demonstrating the two properties are indeed not contradictory [1].  Lastly, we want to emphasize that it is MiDaS, not us, who proposed this loss term. We mainly demonstrate this loss is highly useful (both perceptually and metrically) when applied in the context of high-fidelity synthetic images (L165-167).
>
> [1] Evaluation of CNN-based Single-Image Depth Estimation Methods, ECCV Workshop 2018.

---

> > ### Comment · Reviewer_rTF8 · 2024-08-13
> >
> > Thanks for the response. It would be nice to add some of the discussion in Q2, and regarding the gradient matching loss, to the main paper.

---

> > > ### Author Response · Authors · 2024-08-13
> > >
> > > Thanks for your valuable advice! We will ensure the discussion in Q2 and the discussion on the gradient matching loss are included in our final version. Thank you.

---

### Author Rebuttal · Authors · 2024-08-07

We thank all the reviewers for their constructive and insightful feedback. We have addressed each of the raised concerns individually. Additionally, we have provided a one-page PDF below with further visualizations and qualitative comparisons. We look forward to your further feedback. Thank you very much.

---

### Decision · Program_Chairs · 2024-09-25

**Decision:**

Accept (poster)

**Comment:**

This paper introduces v2 of Depth Anything, which has the following improvements over v1: (1) replaces all labeled real images with accurate synthetic images; (2) expands model capacity by using more powerful backbone models and provide different model sizes for different applications; (3) applies teacher-student models to overcome domain gaps. Both quantitative and qualitative results show that it outperforms the state-of-the-art. The paper received all the positive reviews, with the reviewers appreciating the well-written paper, the impressive results of accurate and detailed monocular depth estimation, and the new benchmark with increased diversity of test scenes that can inspire future work. AC recommends acceptance of this work and strongly encourages the authors to incorporate the rebuttal details into the final camera-ready version as suggested by the reviewers.